# Host phospholipid peroxidation fuels ExoU-dependent cell necrosis and supports *Pseudomonas aeruginosa*-driven pathology

Salimata Bagayoko[1], Stephen Adonai Leon-Icaza[1☢], Miriam Pinilla[1☢], Audrey Hessel[1☢], Karin Santoni[1], David Péricat[1], Pierre-Jean Bordignon[1], Flavie Moreau[1,2], Elif Eren[1], Aurélien Boyancé[1], Emmanuelle Naser[1,3], Lise Lefèvre[4], Céline Berrone[1,2], Nino Iakobachvili[5], Arnaud Metais[1], Yoann Rombouts[1], Geanncarlo Lugo-Villarino[1], Agnès Coste[4], Ina Attrée[6], Dara W. Frank[7], Hans Clevers[8], Peter J. Peters[5], Céline Cougoule[1], Rémi Planès[1¤*], Etienne Meunier[1¤*]

1 Institute of Pharmacology and Structural Biology (IPBS), University of Toulouse, CNRS, Toulouse, France, 2 Level 3 Biosafety Animal Core facility, Anexplo platform, Institute of Pharmacology and Structural Biology (IPBS), University of Toulouse, CNRS, Toulouse, France, 3 Cytometry & Imaging Core facility, Institute of Pharmacology and Structural Biology (IPBS), University of Toulouse, CNRS, Toulouse, France, 4 RESTORE institute, University of Toulouse, CNRS, Toulouse, France, 5 Division of Nanoscopy, Maastricht Multimodal Molecular Imaging Institute, Maastricht University, Maastricht, The Netherlands, 6 Univ. Grenoble Alpes, CNRS, CEA, IBS, Bacterial Pathogenesis and Cellular Responses, Grenoble, France, 7 Department of Microbiology and Immunology, Medical College of Wisconsin, Milwaukee, Wisconsin, United States of America, 8 Oncode Institute, Hubrecht Institute, Royal Netherlands Academy of Arts and Sciences and University Medical Center, Utrecht, Netherlands

☢ These authors contributed equally to this work.
¤ Current Address: Institute of Pharmacology and Structural Biology (IPBS), CNRS, Toulouse; France
* remi.planes@ipbs.fr (RP); etienne.meunier@ipbs.fr (EM)

**Data Availability Statement:** All relevant data are within the manuscript and its Supporting Information files S1 and S2 Datas.

## Abstract

Regulated cell necrosis supports immune and anti-infectious strategies of the body; however, dysregulation of these processes drives pathological organ damage. *Pseudomonas aeruginosa* expresses a phospholipase, ExoU that triggers pathological host cell necrosis through a poorly characterized pathway. Here, we investigated the molecular and cellular mechanisms of ExoU-mediated necrosis. We show that cellular peroxidised phospholipids enhance ExoU phospholipase activity, which drives necrosis of immune and non-immune cells. Conversely, both the endogenous lipid peroxidation regulator GPX4 and the pharmacological inhibition of lipid peroxidation delay ExoU-dependent cell necrosis and improve bacterial elimination *in vitro* and *in vivo*. Our findings also pertain to the ExoU-related phospholipase from the bacterial pathogen *Burkholderia thailandensis*, suggesting that exploitation of peroxidised phospholipids might be a conserved virulence mechanism among various microbial phospholipases. Overall, our results identify an original lipid peroxidation-based virulence mechanism as a strong contributor of microbial phospholipase-driven pathology.

**Funding:** This project was funded by grants from the National Research Agency (ANR, Endiabac), FRM "Amorçage Jeunes Equipes" (AJE20151034460), ERC StG (INFLAME 804249) and ATIP-Avenir program to EM, from National Research Agency (ANR, MacGlycoTB) to YR, from the European Society of Clinical Microbiology and Infectious Diseases (ESCMID, 2020) to RP, from the Van Gogh Programme to IPBS-M4i institutes, from Invivogen-CIFRE collaborative PhD fellowship to MP and from the FRM (FDT202106012794), Mali and Campus France cooperative agencies to SB. The funders had no role in study design, data collection and analysis, decision to publish, or preparation of the manuscript.

**Competing interests:** The authors have declared that no competing interests exist.

## Author summary

Although a proper activation of various regulated cell necrosis confer a significant advantage against various infectious agents, their dysregulation drives host tissue damages that can end up with fatal sepsis. Specifically, 30% of the bacterial strains of *Pseudomonas aeruginosa* (*P. aeruginosa*) express the phospholipase A2-like toxin ExoU that is injected into host target cells through the Type-3 Secretion System. This toxin induces, through a yet unknown mechanism, a strong and fast necrotic cell death that supports fatal respiratory infections. Therefore, in this study, we sought to determine the cellular mechanisms by which ExoU triggers host cell necrosis. In this context, we found that ExoU exploits basal cellular phospholipid peroxidation to promote cell necrosis. Mechanistically, host cell lipid peroxidation stimulates ExoU phospholipase activity, which then triggers a pathological cell necrosis both *in vitro* and *in vivo*. Altogether, our results unveil that targeting host cell lipid peroxidation constitutes a virulence mechanism developed by microbial phospholipases, a process that contributes to *P. aeruginosa*-mediated pathology.

## Introduction

Regulated cell necrosis (RCNs) drives physiological and immune processes, yet dysregulation of this process promotes pathological responses such as organ-failure and sepsis [1–4]. Mechanistically, oxygen-dependent cell death is an evolutionary conserved process that involves the production of reactive oxygen species (ROS), transition metals (e.g. iron) and peroxidised lipid accumulation [5–8]. In addition to cell necrosis, lipid peroxidation broadly involves cellular processes essential to mediate optimal efferocytosis of dead cells, cellular communication resulting from the formation of lipids derived from peroxided phospholipids (e.g. isoprostanes, platelet activating factor) or the production of bioactive lipids (eicosanoids) from arachidonic acid [9,10]. In addition, the peroxidation of the mitochondrial phospholipid cardiolipin initiates apoptosis while the accumulation of peroxidised phosphatidyl ethanolamines (PE) promote the cellular necrosis, ferroptosis [11–17]. Specifically, the dysregulation of lipid peroxidation processes is associated with various human pathologies such as cancer chemoresistance, brain and ischemia injuries, neurological alterations, metabolic diseases as well as tuberculosis susceptibility [18–23]. In this context, the enzymes glutathione peroxidase 4 (GPX4) and ferroptosis-suppressor protein-1 (FSP1) that belongs to the CoQ antioxidant system, detoxify phospholipid hydroperoxide accumulation, hence allowing lipid peroxide amounts to be balanced in cells [5,11,12,14,24]. On the contrary, iron excess, lipoxygenase activity or cytochrome P450 oxidoreductase (CYPOR) all promote phospholipid peroxidation, which can end with ferroptosis induction in the absence of proper regulation [5,14–16,25,26].

In this regard, the bacterial pathogen *Pseudomonas aeruginosa (P. aeruginosa)* expresses ExoU, an A2 phospholipase from the patatin family, that triggers a necrosis-dependent pathology through a poorly understood pathway [27–36]. In presence of cellular co-factors such as ubiquitin [31] or the trafficking chaperone DNAJC5 [37], ExoU activity rapidly cleaves at the sn-2 position of host membrane phospholipids, liberating large amounts of arachidonic acid that are then metabolized into eicosanoids by cellular enzymes cyclooxygenases, cytochrome P450 or lipoxygenases [32,38–40]. Importantly, *in vivo*, ExoU expression by *P. aeruginosa* is associated with a robust production of oxidized lipids such the platelet activating factor (PAF) or isoprostanes [38,41]. In this context, we explored the possibility that *P. aeruginosa* ExoU mediates a necrosis-dependent host pathology involving lipid peroxidation.

## Results

### *P. aeruginosa* infection triggers ExoU-dependent alarmin and peroxidised lipid production in mice

*P. aeruginosa* ExoU is injected into cells by the Type-3 Secretion System (T3SS) [28,36], which triggers a fast and violent cellular necrosis. Therefore, we first monitored the profile of ExoU-dependent pathology in mice infected with the clinical isolate PP34 *exoU⁺* or its isogenic mutant (*exoU⁻*). Similar to previous studies [27,29,42,43], intranasal instillation with either *exoU⁺* or *exoU⁻* strains highlighted a *P. aeruginosa*-induced acute pathology mainly due to ExoU, as mice infected with *exoU⁻* bacteria showed improved survival to infection (**Fig 1A**). This observation was paralleled with lower bacterial loads of *P. aeruginosa exoU⁻* than *exoU⁺* in the bronchoalveaolar lavage fluids (BALFs), the lungs, the blood and the spleen, suggesting that ExoU also promotes bacterial dissemination (**Fig 1B**). As *P. aeruginosa* triggers NLRC4-, NLRP3- and Caspase-11-dependent inflammasome response [42,44,45–52,53], we infected inflammasome-deficient mice (*Casp1/Casp11⁻ᐟ⁻*, *Nlrc4⁻ᐟ⁻* and *GasderminD⁻ᐟ⁻*) and observed that those mice were not protected against *P. aeruginosa exoU⁺*, hence suggesting that ExoU-promoted mouse pathology occurs independently from the inflammasome machineries (**S1A and S1B Fig**). A hallmark of host cell necrosis is the release of intracellular mediators such as alarmins that contribute to the initiation and the development of an inflammatory reaction, which occurs upon *P. aeruginosa* infection [42,54]. Therefore, we primarily focused our analysis on alarmin release. We observed a strong ExoU-dependent alarmin production in BALFs 6

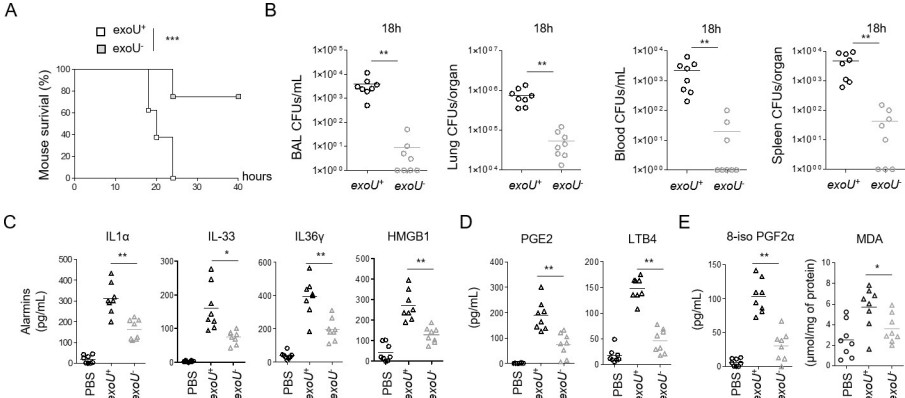

**Fig 1. ExoU-dependent lung pathology in mice associates to an alarmin and peroxidized lipid signature. (A)** Survival of WT mice intranasally infected (n = 7 animals per condition) with 5.10⁵ CFUs of *P. aeruginosa* PP34 or its isogenic mutant PP34*exoU⁻.* Graphs represent one experiment (8 mice/group) out of three independent *in vivo* experiments. Log-rank Cox-Mantel test was used for survival comparisons. ***p ≤ 0.001. **(B)** Bronchoalveolar (BAL), lung, blood and spleen bacterial loads from WT mice (n = 8) 18 hours after intranasal infection with 5.10⁵ CFUs of *P. aeruginosa* PP34 or its isogenic mutant PP34*exoU⁻*. Graphs represent one experiment (8 mice/group) out of three independent *in vivo* experiments. **p ≤ 0.01, Mann-Whitney analysis test. **(C)** Alarmin levels in bronchoalveolar fluids (BALFs) from WT mice (n = 8) 6 hours after intranasal infection with 5.10⁵ CFUs of *P. aeruginosa* PP34 or its isogenic mutant PP34*exoU⁻*. Graphs represent one experiment (8 mice/group) out of three independent *in vivo* experiments; *p ≤ 0.05; **p ≤ 0.01, Mann-Whitney analysis test. **(D)** Prostaglandin E2 (PGE2) and Leukotriene B4 (LTB4) levels in bronchoalveolar fluids (BALFs) from WT mice (n = 8) 6 hours after intranasal infection with 5.10⁵ CFUs of *P. aeruginosa* PP34 or its isogenic mutant PP34*exoU⁻*. Graphs represent one experiment (8 mice/group) out of three independent *in vivo* experiments; **p ≤ 0.01, Mann-Whitney analysis test. **(E)** Peroxidized lipid product (isoprostanes and MDA) levels in bronchoalveolar fluids (BALFs) from WT mice (n = 8) 6 hours after intranasal infection with 5.10⁵ CFUs of *P. aeruginosa* PP34 or its isogenic mutant PP34*exoU⁻*. Graphs represent one experiment (8 mice/group) out of three independent *in vivo* experiments; *p ≤ 0.05; **p ≤ 0.01, Mann-Whitney analysis test. Data information: Data shown as means (**Graphs B-E**) and are representative of one experiment performed three times; *p ≤ 0.05; **p ≤ 0.01, ***p ≤ 0.001, Mann-Whitney analysis test (**B-E**) and log-rank Cox-Mantel test for survival comparisons (**A**).

h after infection, such as IL-1 family alarmins IL1α, IL-33 or IL-36γ [55] (**Fig 1C**). In addition, we also detected that *exoU*-expressing *P. aeruginosa* triggered a strong production of phospholipid- and arachidonic acid (aa)-derived mediators such as prostaglandin E2 and leukotriene B4, which correlates with the robust phospholipase activity of ExoU (**Fig 1D**) [38–40]. Importantly, BALFs of mice infected with *exoU*-expressing *P. aeruginosa* also exhibited a marked presence of oxidized lipid (by)-products such as isoprostanes (8-iso PGF2α) or Malondialdehyde (MDA), which suggests that *exoU*-expressing *P. aeruginosa* also drives an exacerbated lipid oxidation response in mice (**Fig 1E**) [41,56].

## Lipid peroxidation contributes to ExoU-induced cell necrosis and *P. aeruginosa* escape from phagocyte-mediated killing

The observation that *exoU*-expressing *P. aeruginosa* infection associates to a lipid peroxidation signature *in vivo*, encouraged us to determine the importance of lipid peroxidation on ExoU-induced cellular necrosis. As *P. aeruginosa* strains that do not express ExoU can promote an NLRC4 inflammasome response in macrophages [50], we used mouse Bone-Marrow-Derived Macrophages (BMDMs) that lack *Nlrc4* expression to specifically address the importance of lipid peroxidation on ExoU-dependent cell necrosis. We infected *Nlrc4*⁻/⁻ primary murine BMDMs with *P. aeruginosa* strains expressing or not expressing ExoU. The pharmacological inhibition of various regulated necrosis pathways (e.g. pyroptosis, necroptosis, apoptosis, parthanatos) showed that only ferrostatin-1, a potent and well characterized inhibitor of phospholipid peroxidation [57], repressed ExoU-dependent cell necrosis (**Figs 2A and 2B and S2A and S1–S6 Movies**). Ferrostatin-1 action was specific to lipid peroxidation-dependent cell necrosis as it also inhibited Cumene hydroperoxide-induced ferroptosis (CuOOH, 400μM) but not Flagellin-/LPS-induced pyroptosis or TCPA-1/Z-VAD/TNFα-dependent necroptosis (**Fig 2B**). In addition, ExoU-induced IL-1α and HMGB1 alarmin release in macrophages was reduced in presence of ferrostatin-1 whereas TNFα levels remained similar (**Fig 2C**), suggesting that lipid peroxidation contributes to alarmin release in response to ExoU. We noticed that ExoU-triggered ferrostatin-1-sensitive necrosis was not restricted to murine BMDMs as primary human macrophages, the human U937 monocytic cell line, human and murine neutrophils and eosinophils, the human bronchial epithelial (HBEs), A549 or HeLa epithelial cells were all sensitive to lipid peroxidation inhibition upon infection with *exoU*-expressing *P. aeruginosa* (**S2A and S2B Fig**). ExoU exhibits a calcium-independent phospholipase A2-like activity [33]. Hence, we transfected recombinant ExoU protein (rExoU) or its enzymatically inactive mutant ExoU^S142A [58] in WT BMDMs and monitored for cell necrosis. Only macrophages transfected with active ExoU underwent to cell death, a process that was inhibited by the use of ferrostatin-1 or the phospholipase inhibitor MAFP (**S2C Fig**). In line, we found that ferrostatin-1 itself did not alter bacterial growth or ExoU secretion (**S2D and S2E Fig**), suggesting that ferrostatin-1 does not directly alter bacterial physiology nor expression/secretion of ExoU. Upon phospholipase activation arachidonic acid release can be metabolized and oxidized by various cellular enzymes, including cyclooxygenases 1 and 2 (COX1, COX2), lipoxygenases (ALOX5 and ALOX12/15 in mice) or cytochrome p450 (CYPs) enzymes. Therefore, we transfected recombinant ExoU in WT, *Alox5*⁻/⁻ or *Alox12/15*⁻/⁻ BMDMs in presence or absence of various COX, CYP or different lipid peroxidation inhibitors (a-tocopherol, liproxstatin-1, Resveratrol, ferrostatin-1). Although we observed that all lipid peroxidation inhibitors have a strong inhibitory impact on cell death, cyclooxygenase, cytochrome P450 or lipoxygenase targeting did not interfere with ExoU-dependent cell necrosis, hence suggesting that those enzymes do not regulate lipid-peroxidation-dependent cell necrosis upon ExoU exposure (**Fig 2D**). Importantly, we also observed that ferrostatin-1 delayed ExoU-induced cell necrosis,

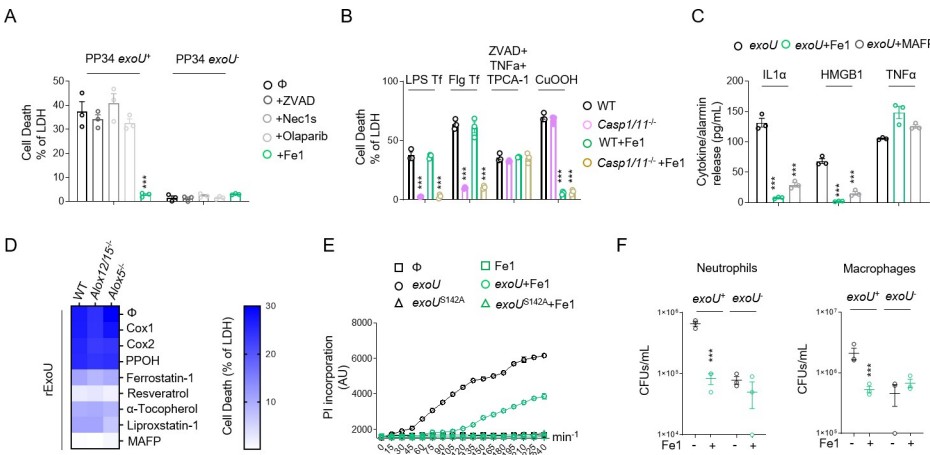

**Fig 2. Lipid peroxidation inhibition delays ExoU-induced cell necrosis.** Otherwise specified, cells were infected with an MOI of 0.5 of *P. aeruginosa* PP34, PP34$^{exoU-}$ or PP34$^{exoUS142A}$ for various times. ***p ≤ 0.001, t-test with Bonferroni correction. **(A)** Measure of LDH release in *Nlrc4$^{-/-}$* BMDMs infected with PP34 or PP34$^{exoU-}$ in presence of Z-VAD (40μM), olaparaib (10μM), Necrostatin-1s (Ne1s, 40μM) or Ferrostatin-1 (Fe1, 10μM) for 2 hours. ***p ≤ 0.001, t-test with Bonferroni correction. **(B)** Measure of LDH release in WT or *Casp1$^{-/-}$/Casp11$^{-/-}$* BMDMs transfected (Lipofectamin 2000) with 1μg of LPS or Flagellin (Flg) to induce pyroptosis, treated with Z-VAD (40μM)/TNFα (500UI/mL)/TPCA-1 (5 μM) to induce necroptosis or with Cumene hydroperoxide (CuOOH, 400μM) to induce ferroptosis in presence or absence of Ferrostatin-1 (Fe1, 10μM) for 6 hours. ***p ≤ 0.001, t-test with Bonferroni correction. **(C)** Measure of alarmin/cytokine release in *Nlrc4$^{-/-}$* BMDMs infected with PP34 or PP34$^{exoU-}$ in presence of Z-VAD (40μM), olaparaib (10μM), Necrostatin-1s (Ne1s, 40μM) or Ferrostatin-1 (Fe1, 10μM) for 2 hours. ***p ≤ 0.001, t-test with Bonferroni correction. **(D)** Heat map representing measure of LDH release in WT, *ALOX5$^{-/-}$* and *ALOX12/15* BMDMs transfected with recombinant ExoU in presence/absence of Cox1 inhibitor (Ketorolac Tromethamine, 10μM), Cox2 inhibitor (NS 398, 25μM), Cyp450 epoxygenase activity inhibitor (PPOH, 10μM), phospholipase inhibitor MAFP (50μM) or lipid peroxidation inhibitors Ferrostatin-1 (Fe1, 20μM), Resveratrol (5μM), Liproxstatin-1 (30μM), a-Tocopherol (20μM) for 2 hours. The heat map shows the mean of three combined independent experiments, each performed in triplicate. **(E)** Time course measure of plasma membrane permeabilization using propidium iodide incorporation in *Nlrc4$^{-/-}$* BMDMs infected with PP34 or PP34$^{exoUS142A}$ in presence/absence of Ferrostatin-1 (Fe1, 20μM). ***p ≤ 0.001, t-test with Bonferroni correction. **(F)** Microbicidal activity of macrophages (5h) and neutrophils (3h) after infection with *P. aeruginosa exoU$^+$* and *exoU$^-$* (MOI 0.5) in presence/absence of ferrostatin-1 (10μM). ***p ≤ 0.001, t-test with Bonferroni correction. Data information: Data are represented as means +/- SEM (graphs A- F) from n = 3 independent pooled experiments; ***P ≤ 0.001 for the indicated comparisons using t-test with Bonferroni correction.

suggesting either that the phospholipase activity of ExoU promotes lipid peroxidation-independent cell death or that the inhibitory effect of ferrostatin-1 is unstable over time (**Fig 2E**). Regarding this, the replenishment of *P. aeruginosa*-infected cells with fresh ferrostatin-1 each hour strongly improved cell viability, suggesting that the instability of Fe1 might also account in the delayed ExoU-induced cell necrosis we observed (**S2F Fig**). Finally, we evaluated if the inhibition of lipid peroxidation would modulate macrophage and neutrophil microbicidal response upon *exoU*-expressing *P. aeruginosa* infection. We observed that ferrostatin-1 strongly improved both macrophage and neutrophil microbicidal activities to a level close to those observed in response to *exoU*-deficient *P. aeruginosa* (**Fig 2F**), hence suggesting that *P. aeruginosa* ExoU relies on lipid peroxidation-dependent cell necrosis to escape from phagocyte attack. Together, our results suggest that host cell lipid peroxidation is important for ExoU-induced host cell necrosis and release of alarmins.

## Lipid peroxidation fuels ExoU phospholipase activity

Lipid-peroxidation requires reactive oxygen species (ROS), such as $H_2O_2$, that can oxidize various phospholipids [5]. Therefore, we evaluated the ability of ExoU to induce ROS-dependent lipid peroxidation in macrophages. Although we observed that, 30 minutes after transfection,

ExoU but not its catalytically inactivated mutant ExoU[S142A], triggered an acute ROS production in BMDMs, we surprisingly failed to detect a robust lipid peroxidation accumulation as measured by the C11 Bodipy probe (**Figs 3A and S3A and S3B**). As control, the well-known lipid peroxidation inducer Cumene hydroperoxide (CuOOH) promoted cellular lipid peroxidation (**Fig 3A**) [59]. In contrast, we observed that basal lipid peroxidation in cells was reduced upon ExoU transfection or PP34 infection, a process that was further strengthened in presence of ferrostatin-1 (**Figs 3A and S3B**).

These results suggest that, instead of promoting pathological lipid peroxidation, ExoU might actually use cellular lipid peroxidation to promote cell necrosis. To this regard, various host phospholipase A2 enzymes have been described to specifically cleave and remove peroxidised phospholipids from membranes [60–62]. To address this hypothesis, we performed a redox phospholipidomic approach to determine if ExoU could interfere with the endogenous levels of peroxidised phospholipids (**Figs 3B and S3C**). We used a 45 min time-point to perform our experiments, as a point where plasma membrane permeabilization (propidium uptake monitoring) is not observed. This design excludes the possibility that a decrease in peroxidised phospholipids is due to cell necrosis induced by ExoU (**S3D Fig**). We observed that ExoU-treated macrophages had a decrease in peroxidised phospholipids as measured by the reduction in hydroperoxil (-OOH)- and hydroxyl (-OH)-phosphoinositols (PIs)/- phosphoserines (PSs) and—phosphocholines (PCs) with arachidonic acid (C20:4/C22:4) acid side chains (**Figs 3B and S3C**).

In cells, peroxidised phospholipids are detoxified by various factors, one of the most important being the ferroptosis regulator glutathione peroxidase 4 (GPX4) [5]. Consequently, the use of pro oxidant molecules or *Gpx4* genetic inactivation both induce a strong accumulation of various peroxidised phospholipids in cell membranes [5]. Therefore, we hypothesized that prestimulation of macrophages with non-cytotoxic doses of the lipid peroxidation and ferroptosis inducer Cumene hydroperoxide (20µM, 1h) might sensitize cells to ExoU-induced cell necrosis. We transfected recombinant (r)ExoU in WT BMDMs in presence or absence of nontoxic doses of the pro-oxidant Cumen hydroperoxide (CuOOH, 20µM, 1h) [59]. Although CuOOH promoted lipid peroxidation but not BMDM cell death, rExoU transfection specifically induced an increased cell necrosis in CuOOH-primed BMDMs, a process that was inhibited by the use of ferrostatin-1 (**Fig 3C and 3D**). In agreement with this result, we measured a strong decrease in lipid peroxidation in CuOOH-primed cells transfected with rExoU (**Fig 3C and 3D**), confirming that ExoU efficiently targeted lipid peroxides induced by CuOOH. In addition, microscopy observations of CuOOH-primed cells highlighted a decrease of peroxidized lipids at the plasma membrane upon infection by ExoU-expressing strain of *P. aeruginosa* (PP34), suggesting that ExoU mostly target plasma membrane peroxidized phospholipids to promote cell necrosis (**Fig 3E**). The enzyme cytochrome p450 oxidoreductase (CYPOR) has recently been found to be an important provider of peroxidized phospholipids upon ferroptosis induction; we hypothesized that ExoU function might be regulated by CYPOR-regulated phospholipid peroxidation. We acquired *Cypor*-deficient HeLa cells but also generated *Cypor*[-/-] immortalized (i)BMDMs using CRISPR (**S3E Fig**) and evaluated the importance of CYPOR on ExoU-driven cell necrosis. PP34 infection of WT and *Cypor*[-/-] immortalized BMDMs triggered similar cell deaths, suggesting that in resting cells, CYPOR does not promote the basal lipid peroxidation involved in ExoU-dependent cell necrosis (**Fig 3G and 3H**). However, in CuOOH-primed macrophages, where phospholipid peroxidation is induced, we observed that CYPOR was a major contributor of phospholipid peroxidation (**S3F Fig**). This was associated to enhanced ability of PP34 to trigger cell necrosis in CuOOH-primed WT but not in *Cypor*[-/-] iBMDMs and HeLa cells (**Fig 3G and 3H**), which suggests that CYPOR-induced lipid peroxidation heightens ExoU-dependent toxicity. However, in resting cells,

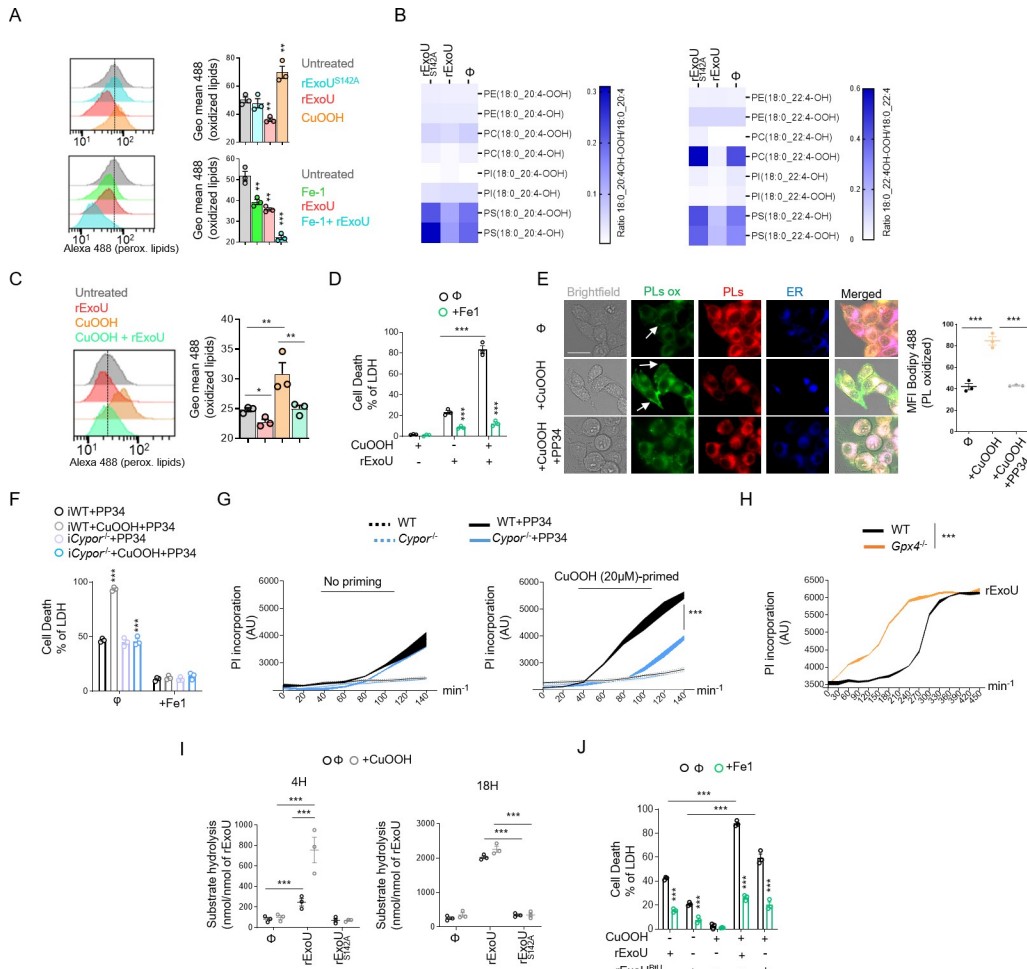

**Fig 3. ExoU-induced cell death involves ROS-induced lipid peroxidation but proceeds in a ferroptosis independent manner.** (A) Cytometry detection and quantification of (phospho)lipid peroxidation using the probe C11-bodipy in WT BMDMs treated with CuOOH (20µM) or transfected with rExoU (500ng) or its catalytically dead mutant rExoU$^{S142A}$ (500ng) for 1 hour in presence or absence of Ferrostatin-1 (20µM). Sample were acquired using FACSCalibur (BD). The graphs shows the mean+/-SEM of one experiment performed in triplicate out of three independent experiments. **P≤0.001, ***P ≤ 0.001 for the indicated comparisons using t-test with Bonferroni correction. (B) (Redox) lipidomic analysis of phospholipid peroxidation in BMDMs transfected with recombinant ExoU or its catalytically dead mutant ExoU$^{S142A}$ for 45 minutes. Each value is standardized to the corresponding phospholipid content shown in (S3B Fig). The heat map shows the mean of one experiment performed in triplicate. (C) Cytometry detection and quantification of (phosphor)lipid peroxidation using the probe C11-bodipy in WT BMDMs pre-treated or not for 1 hour with CuOOH (20µM) in presence or absence of Ferrostatin-1 (20µM) and then transfected with rExoU (500ng) for 1 hour. Sample were acquired using FACSCalibur (BD). The graphs shows the mean+/-SEM of one experiment performed in triplicate out of three independent experiments. *P ≤ 0.05, **P≤0.001, for the indicated comparisons using t-test with Bonferroni correction. (D) Measure of LDH release in WT BMDMs pre-treated or not for 1 hour with CuOOH (20µM) in presence or absence of Ferrostatin-1 (20µM) and then transfected with rExoU (500ng) for 3 hours. ***p ≤ 0.001, T-test with Bonferroni correction. (E) Representative microscopy images (phospho)lipid peroxidation and quantifications using the probe C11-bodipy in CuOOH-primed (20µM) HELA cells infected with PP34 (MOI5) for 2 hours. Images show two independent experiments, each performed three times at 2 hours post infection. Scale bar 20µm; Green, oxidized bodipy (oxidized phospholipids, PLs ox); Red, bodipy (phospholipids, PLs); Blue (Endoplasmic Reticulum, ER tracker probe, 1µM). Arrows show enriched peroxidised phospholipids in the plasma membrane area. Quantifications show the Mean Fluorescence Intensity (MFI) quantification of Peroxidized lipids from one experiment performed three times (50–60 cells counted). ***P<0.001 by T-test. (F) Measure of LDH release in immortalized (i) WT or *Cypor*$^{-/-}$ BMDMs primed or not with CuOOH (20µM, 1hour) in presence or absence of ferrostatin-1 (20µM) and infected for 2 hours with PP34. ***p ≤ 0.001, T-test with Bonferroni correction. (G) Time course measure of plasma membrane permeabilization using propidium iodide incorporation in WT and *Cypor*$^{-/-}$ HELA cells primed or not with CuOOH (20µM, 1hour) and infected with PP34 (MOI5) for 2 hours. ***p ≤ 0.001, T-test with Bonferroni correction. (H) Time course measure of plasma membrane permeabilization using propidium iodide incorporation in immortalised WT and *Gpx4*$^{-/-}$ BMDMs transfected with rExoU (500ng) for 7 hours. ***p ≤ 0.001, T-test with Bonferroni correction. (I) ExoU phospholipase

activity determination in WT BMDM lysates pre-treated or not with CuOOH (20μM, 1hour). 100 pmols of ExoU were used and phospholipase hydrolysis rate (nmoles of substrate hydrolysed/nmole of ExoU) was measured after 4 h and 16 hours. ***p ≤ 0.001, T-test with Bonferroni correction. **(J)** Measure of LDH release in WT BMDMs primed or not with CuOOH (20μM, 1hour) in presence or absence of ferrostatin-1 (20μM) and transfected for 3 hours with 5μg of rExoU[BtU] or 500ng rExoU. ***p ≤ 0.001, T-test with Bonferroni correction. Data information: Data are plotted as means+/- SEM **(D, F-J)** from n = 3 independent pooled experiments; ***P ≤ 0.001 for the indicated comparisons using t-test with Bonferroni correction.

basal lipid peroxidation appears to be regulated by other processes/enzymes. Finally, using Crispr-Cas9, we generated *Gpx4*[-/-] immortalized BMDMs **(S3G Fig)**. As previously observed by others in other cell lines [16,24], *Gpx4*[-/-] immortalized BMDMs exhibited increased basal levels of peroxidised lipids **(S3H Fig)**. Therefore, rExoU transfection triggered faster cell death of *Gpx4*[-/-] macrophages than their WT counterpart, suggesting that lipid peroxidation of cells enhances ExoU-dependent toxicity **(Figs 3H and S3G and S3H)**.

Upon phospholipid peroxidation, arachidonic acid-containing phospholipids form isoprostanes that are potent intra- and extra-cellular inflammatory mediators [9,10]. Once formed, these isoprostanes are released from phospholipids by the action of phospholipases [9,10]. Therefore, we reasoned that if ExoU targets peroxidised phospholipids, this would promote ExoU phospholipase-dependent release of endogenous pre-formed isoprostanes. Accordingly, the release of the 8-PGF2α isoprostane was specifically induced by ExoU in WT macrophages, a process that was further amplified by the co treatment of cells with non-toxic concentrations of Cumen hydroperoxide (CuOOH 20μM, 1 h) and ExoU **(S3I Fig)**. Of importance, ferrostatin-1 strongly inhibited ExoU- and ExoU/CuOOH-induced 8-PGF2α release **(S3I Fig)**. In addition, we also detected that in CuOOH-primed macrophages, the amount of arachidonic acid-derived eicosanoids leukotriene B4 and prostaglandin E2, which are an indirect indication of the phospholipase activity of ExoU, were also strongly increased after the exposure to ExoU, hence suggesting that ExoU-targeted peroxidised phospholipids might increase its phospholipase activity toward all phospholipids (peroxidized or not) **(S3J Fig)**. Consequently, we measured the phospholipase activity of ExoU in cell lysates where we chemically induced non-lethal lipid peroxidation with Cumene hydroperoxide (CuOOH, 20μM) for 1 h or not. We observed that in CuOOH-primed cell lysates, ExoU exhibited a stronger activity than in unprimed samples after 4 h of incubation **(Fig 3I)**. Importantly, after 18 h incubation, we observed the same accumulation of hydrolysed substrate in CuOOH-primed and unprimed samples, which suggests that lipid peroxidation exacerbates the early activation of ExoU **(Fig 3I)**. As control, ExoU[S142A]- treated cell lysates did not show a significant phospholipase activity induction, suggesting that we mostly measured the PLA2 activity from ExoU, but not from cellular phospholipases **(Fig 3I)**. Finally, we aimed at challenging our findings by determining if other toxic phospholipases also had a similar activation pattern to ExoU. Hence, we transfected macrophages with the closely related patatin-like phospholipase A2 from *Burkholderia thailandensis* (ExoU[BtU]) [31]. We observed that recombinant ExoU[BtU] transfection induced BMDMs necrosis, a process that was exacerbated by CuOOH priming and inhibited by the use of ferrostatin-1, suggesting that ExoU[BtU] also follows a pattern involving host cell lipid peroxidation **(Fig 3J)**. Altogether, our results suggest a surprising mechanism by which ExoU exploits cellular lipid peroxidation to trigger necrosis, a process that can be extended to the action of *B. thailandensis* ExoU[BtU]-related phospholipase.

## Ferrostatin-1 improves mouse resistance to infection by *exoU*-expressing *P. aeruginosa*

ExoU-induced necrosis promotes host lung pathology, which leads to a sepsis like response as well as respiratory failure syndrome. Therefore, we hypothesized that ferrostatin-1 use could

protect mice against *exoU*-expressing *P. aeruginosa*. Intranasal infection of mice using *P. aeruginosa exoU*[+] showed that mice intraperitoneally pre-treated with ferrostatin-1 (6 h before infection, 6mg.k[-1]) had diminished bacterial loads in BALFs, lungs and spleen. Ferrostatin-1 pre-treatment did not significantly modify bacterial loads of *exoU*-deficient bacteria, suggesting that ferrostatin-1 mainly modulates ExoU-dependent processes in mice (**Fig 4A**). Similarly, ferrostatin-1 also attenuated ExoU-dependent alarmin release (e.g. IL-36γ, IL33, IL1α) and the level of oxidized lipids (isoprostanes, MDA) in the BALs (**Fig 4B and 4C**). Additionally, evaluation of the cellular contents in BALFs showed that ferrostatin-1 significantly protected a pool of alveolar macrophage upon *P. aeruginosa* challenge simultaneously decreasing the number of recruited neutrophils, eosinophils and monocytes (**Figs 4D and S4A**). Although a pathological function of recruited immune cells such as neutrophils is probable, we hypothesize that ferrostatin-1-inhibited resident alveolar macrophage death in response to *exoU*-expressing *P. aeruginosa* might confer an improved immune protection characterized by lower immune cell recruitment and lower tissue damages. Regarding this, lung histological observations showed that the inflammatory status of mice infected with non-lethal doses of ExoU-expressing *P. aeruginosa* (1.10[5] CFUs) was improved in presence of ferrostatin-1 (**Fig 4E**). Next, we addressed survival upon ExoU-expressing *P. aeruginosa* challenge. We observed that ferrostatin-1-treated mice (4–6 h before infection, 6mg.k[-1]) had an improved survival rate than those treated with PBS after 40 h after infection (**Fig 4F**). We validated that ferrostatin-1 specifically protected mice against ExoU-induced pathology as ferrostatin-1-treated mice did not show enhanced protection (survival) against ExoU-deficient *P. aeruginosa* (**Fig 4F**).

Finally, we aimed to evaluate if *P. aeruginosa* ExoU would trigger pathological lipid peroxidation-dependent cell necrosis in human bronchial organoids. Organoids were derived from normal lung tissue adjacent to tumors obtained from patients undergoing lung resection due to non-small cell lung carcinoma (NSCLC). Live cell imaging of organoids microinjected with *P. aeruginosa* showed that ExoU triggered complete organoid collapse (**Fig 4G and S7–S12 Movies**). Importantly, ferrostatin-1 strongly attenuated *P. aeruginosa*-dependent organoid damages (**Fig 4G and S7–S12 Movies**). Altogether, our results identified that *P. aeruginosa* ExoU phospholipase benefits from lipid peroxidation to trigger pathology both in mice and in human bronchial organoids.

## Discussion

As a preferential extracellular pathogen, *P. aeruginosa* uses its Type 3-Secretion System (T3SS) to inject virulence factors (Exo S, T, Y and U), allowing bacterial escape from phagocytic uptake and killing. Although *exoS*-expressing *P. aeruginosa* strains associate to the development of chronic infections, *exoU*-expressing *P. aeruginosa* triggers acute deadly infections that associate with a strong oxidative imbalance. In this study, we describe that endogenous basal lipid peroxidation contributes to ExoU-dependent cellular toxicity and mouse pathology. Though we do not exclude that *in vivo*, lipid peroxidation might play various pathological roles that go beyond the sole regulation of cell necrosis, such processes appear to be linked to ExoU expression. In this context, previous studies showed that ExoU promotes production of the platelet-activating factor or the 8-PGF2α isoprostane, two oxidized lipids [41]. In addition, ExoU directly promotes a strong release of arachidonic acid from phospholipids. Enzymes such as cytochrome P450/COXs/LOXs can enzymatically produce oxygenated arachidonic products such as prostaglandin E2/leukotriene B4 involved in pathological signalling pathways upon *P. aeruginosa* infection [38,40,63]. However, results from others and ours mostly suggest that, taken individually, those enzymes only play a negligible role in ExoU-induced cell necrosis [38,40,63]. Regarding the central cell types involved in ExoU-induced pathology, previous studies identified macrophages and

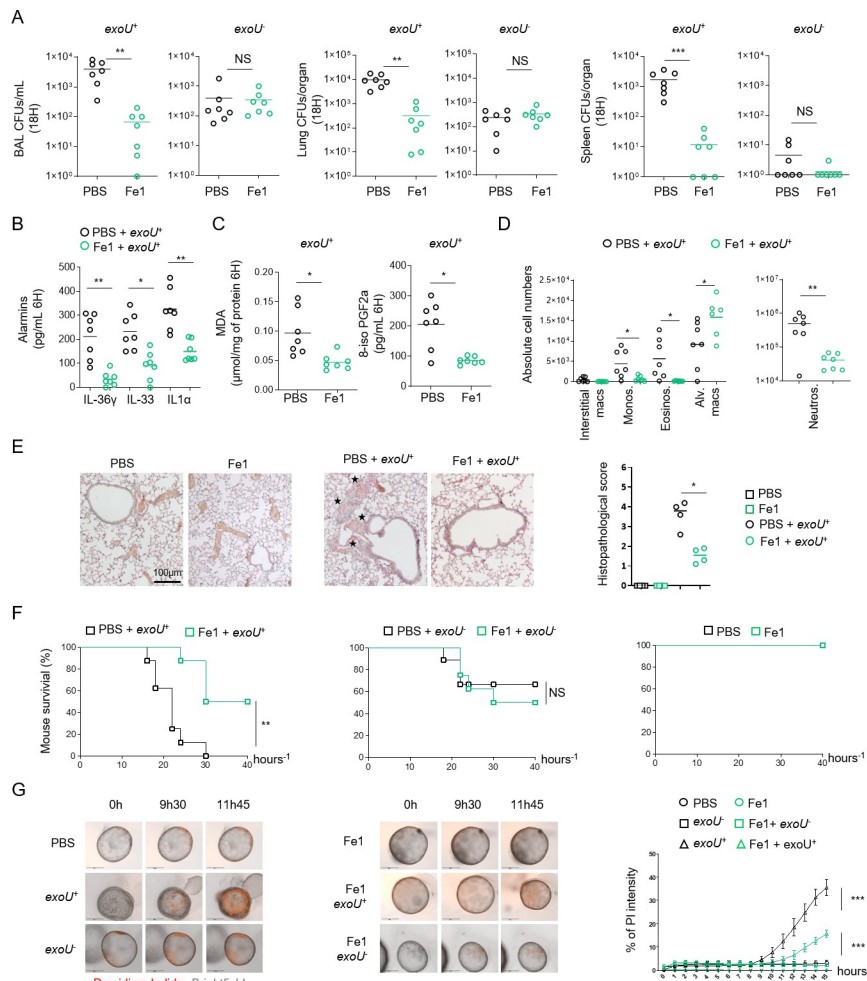

**Fig 4. Ferrostatin-1 protects mice against ExoU-induced lung pathology. (A)** Bronchoalveolar (BAL), lung and spleen bacterial loads from WT mice (n = 7/group) 18 hours after intranasal infection with $5.10^5$ CFUs of *P. aeruginosa* PP34 or its isogenic mutant PP34$^{exoU-}$. When specified, mice were intraperitoneally pretreated with ferrostatin-1 (6mg.k$^{-1}$ or PBS) 4 hours before intranasal infection. Graphs represent one experiment (7 mice/group) out of three independent *in vivo* experiments. $^{**}$p $\leq$ 0.01, Mann-Whitney analysis test. NS: Not significant. **(B, C)** Alarmin and lipid peroxide products levels in bronchoalveolar fluids (BALFs) from WT mice (n = 7 mice/group) 6 hours after intranasal infection with $5.10^5$ CFUs of *P. aeruginosa* PP34 or its isogenic mutant PP34$^{exoU-}$. When specified, mice were intraperitoneally pretreated with ferrostatin-1 (6mg.k$^{-1}$ or PBS) 4 hours before intranasal infection. Graphs represent one experiment (7 mice/group) out of three independent *in vivo* experiments; $^{*}$p $\leq$ 0.05; $^{**}$p $\leq$ 0.01, Mann-Whitney analysis test. **(D)** Immune cell (CD45+) populations in bronchoalveolar fluids (BALFs) from WT mice (n = 7 mice/group) 6 hours after intranasal infection with $5.10^5$ CFUs of *P. aeruginosa* PP34 or its isogenic mutant PP34$^{exoU-}$. When specified, mice were intraperitoneally treated with ferrostatin-1 (6mg.k$^{-1}$ or PBS) 4–6 hours before intranasal infection. Graphs represent one experiment (7 mice/group) out of three independent *in vivo* experiments; $^{*}$p $\leq$ 0.05; $^{**}$p $\leq$ 0.01, Mann-Whitney analysis test. **(E)** Histological observation and scoring of bronchial and lung cellular infiltrations upon *exoU*-expressing *P. aeruginosa* intranasal infection. When specified, mice were intraperitoneally pretreated with ferrostatin-1 (6mg.k$^{-1}$ or PBS) 4–6 hours before intranasal infection. Stars show the cellular infiltrates. $^{*}$p $\leq$ 0.05; Mann-Whitney analysis test. **(F)** Mice survival (n = 7 mice/group) 40 hours after intranasal infection with $5.10^5$ CFUs of *P. aeruginosa* PP34 or its isogenic mutant PP34$^{exoU-}$. Mice were intraperitoneally pretreated with ferrostatin-1 (6mg.k$^{-1}$ or PBS) 4 hours before intranasal infection. Graphs represent one experiment (7 mice/group) out of three independent *in vivo* experiments; $^{**}$p $\leq$ 0.01, Log-rank Cox-Mantel test was used for survival comparisons. **(G, H)** Time-lapse microscopy and the associated quantifications of the measure of plasma membrane permeabilization using propidium iodide incorporation in human primary bronchial organoids infected (microinjection) with *P. aeruginosa* expressing *exoU*$^+$ or its isogenic mutant (*exoU*$^-$) in presence or absence of ferrostatin-1 (40μM) for 15 hours. Data are plotted as means+/- SEM. $^{***}$p $\leq$ 0.001, T-test with Bonferroni correction. Data information: Data shown as means (**Graphs A-E**) and are representative of one experiment performed three times; $^{*}$p $\leq$ 0.05; $^{**}$p $\leq$ 0.01, Mann-Whitney analysis test (**A-E**) and log-rank Cox-Mantel test for survival comparisons (**F**).

neutrophils as central targets of ExoU injection by *P. aeruginosa*. Therefore, future studies will be of importance to determine if the respective contribution of each cell type in pathology induced by ExoU-exploited lipid peroxidation. Regarding this, our *in vivo* observations that targeting lipid peroxidation confers some protection of mice against ExoU-dependent pathology is to put in the light of a decrease in some eicosanoid production such as LTB4 and PGE2, two important modulators of ExoU-driven pathology [38,40,63]. Therefore, the use of *Alox5*[−/−] or *Cox*[−/−] mice, unable to generate LTB4 or various prostaglandins respectively, will also help to determine the respective importance of eicosanoid burst or cell necrosis upon lipid peroxidation-driven ExoU activity.

Although controlled phospholipid peroxidation is of importance for the cells to perform various processes such as efferocytosis through the engagement of peroxidised-PS, mitochondria-dependent apoptosis through cardiolipin peroxidation, signal transduction through peroxidised PC-derived lipids, unrestricted accumulation of peroxidised PEs drives ferroptosis [9,10,64]. A question in both basal lipid peroxidation and ferroptosis-induced lipid peroxidation lies on the compartment phospholipid peroxidation occurs. Peroxisiomes are key at providing ether-phospholipids that will be peroxidised upon ferroptosis induction, the Endoplasmic reticulum is also a central regulator of phospholipid turn over and plasma membrane constitutes the probable location of lipid peroxidation-driven cell lysis upon ferroptosis induction [16,65,66]. Our observations also suggest that although lipid peroxidation can occur in various cellular compartments, ExoU-induced cell necrosis mostly arises from plasma membrane cleaved peroxidized phospholipids. Yet, this does not exclude at all that phospholipid peroxidation could occur in other intracellular organelles, including the endoplasmic reticulum.

Ferroptosis is thought to be a constitutively activated form of cell death that is kept under control through the activity of endogenous regulators of lipid peroxidation such as GPX4, FSP1-mediated coQ10 production, $\alpha$-tocopherol (vitamin E). In addition, the host cellular calcium ($Ca^{2+}$)-independent PLA2$\gamma$, the peroriredoxin Prdx6 PLA2 or the PLA2G6 ($Ca^{2+}$-independent PLA2$\beta$) can cleave and remove preferentially peroxidised phospholipids, hence contributing to phospholipid peroxide detoxification [61,62,67–71]. It is important to notice that both the iPLA2beta and iPLA2g belong to the patatin-like phospholipase family, as ExoU, which suggests that this family of phospholipases might have some conserved affinities to peroxidzed phospholipids [72]. The activity of those phospholipases is tightly regulated by various cellular systems (e.g. ROS levels, calcium fluxes, phospholipid composition) that ensure an optimal but not dysregulated phospholipid cleavage [71]. To this regard, our findings that cellular phospholipid peroxidation is a strong enhancer of ExoU-induced pathological necrosis appears in first view counter intuitive. In this context, we envision that, as a virulence factor, ExoU activity does not follow host regulation and uses host peroxided phospholipids to boost its patatin-like A2 phospholipase activity allowing to aberrantly target and cleave host (peroxidised) phospholipids. Consequently, the use of lipid peroxidation inhibitors such as resveratrol, liproxstatin-1 or ferrostatin-1 attenuates the potency and the speed of ExoU-induced cell necrosis. This offers a key time window for macrophage and neutrophil-mediated bacterial uptake and killing. Although, the identification of cellular enzymatic systems that promote basal lipid peroxidation remains to be explored and characterized, lipid peroxidation accumulation upon *Gpx4* removal or oxidant stress enhances ExoU-induced cellular necrosis. It is intriguing that endogenous peroxidised phospholipids favour ExoU-induced cell necrosis, suggesting that ExoU-expressing strains of *P. aeruginosa* take advantage of the host ferroptosis pathways to maximally damage host tissues. Hence, oxidant-activated cytochrome P450 oxidoreductase CYPOR, a crucial regulator of ferroptosis, strongly enhanced ExoU-dependent cell necrosis, which suggests a important link between ferroptosis-regulated pathways and ExoU activity. Should other regulators of ferroptosis such as ACSL and LPCAT acyl transferases on ExoU-dependent toxicity warrants further investigations [15].

Phospholipases are also present in venoms or various microbial pathogens (e.g. *M. tuberculosis*, *L. monocytogenes*, *S. pyogenes*) and can also promote fast cell necrosis [73–75]. Conversely, we extended our findings to the ExoU closely related ExoU$^{BtU}$ phospholipase from *B. thailandensis*. Remarkably, snake, scorpion or spider venoms are a complex mixture of various components, including the L-amino acid oxidase, able to generate $H_2O_2$-driven lipid peroxidation, and secreted phospholipases able to cleave phospholipids [73]. In this context, it is tempting to speculate that venoms have all components necessary to mediate cell damage in a complex single-injection mixture. L-amino acid oxidase-induced lipid peroxidation might work with venom PLA2 to optimize phospholipid cleavage and subsequent cell necrosis. Related to this, Sevanian and colleagues made pioneer observations that the PLA2 activity from the snake *Crotalus adamanteus* is exacerbated in contact of liposomes constituted of peroxidised phospholipids, a process that is thought to be due to the better accessibility of the sn2-peroxidized fatty acid to phospholipase [70]. Whether ExoU and its relatives follow a similar pathway of activation will be studied in future studies.

In a broader point of view, it is interesting to note that phospholipases can promote allergic shock associated with a strong release of the allergic alarmin interleukin-33 [76], a signature we also observed in mice infected with ExoU-expressing *P. aeruginosa*. Should lipid peroxidation be involved in IL33-driven allergy or asthma in response to phospholipases or other allergens (e.g. proteases) [77] will require additional study.

Understanding the mechanisms of regulated cell necrosis and their physio-pathological consequences is currently driving intensive research and debates. While the importance of lipid peroxidation in antigen presentation, anti-cancer treatments or in exacerbating neurodegenerative diseases becomes more and more clear, its function in infectious diseases remains less studied. Regarding this, Dar et al., recently described that, upon chronic infection, secreted *P. aeruginosa* lipoxygenase (loxA) could sensitize cells to lipid peroxidation-induced ferroptosis [22]. In addition, Kain and colleagues recently linked regulation of host lipid peroxidation and ferroptosis to restriction of liver-stage malaria, which suggests that host peroxidised phospholipids might play yet unsuspected functions in immunity or susceptibility to various pathogens [78]. Thus, our findings that the bacterial patatin-like phospholipase A2 ExoU contributes to pathology by exploiting target cell lipid peroxidation adds an additional piece of significance for the role of lipid peroxidation in infectious diseases but also offers novel insights to target host lipid peroxidation pathways in the frame of infectious diseases (**S1 Graphical Abstract**).

## Material and method

### Ethics statements

The use of human cells was performed under the agreement of the Research Ethical Committee, Haute-Garonne, France. Buffy coats came anonymously by the EFS (établissement français du sang, Toulouse, France). For each donor, a written informed consent was obtained according to the EFS contract agreement n˚ 21PLER2017-0035AV02, according, to "Decret N˚ 2007–1220 (articles L1243-4, R1243-61)".

Animal experiments were approved by local (CE01 committee) and national ethic committees (License APAFIS#8521–2017041008135771, Minister of Research, France) and performed according to local guidelines (French ethical laws) and the European Union animal protection directive (Directive 2010/63/EU).

### Mice

*Nlrc4*$^{-/-}$, *Casp1*$^{-/-}$*Casp11*$^{-/-}$, *GsdmD*$^{-/-}$, *ALOX12/15*$^{-/-}$ and *ALOX5*$^{-/-}$ mice were generated and described in previous studies [79–82]. Mice were bred at the IPBS (Toulouse, France) animal

facilities in agreement to the EU and French directives on animal welfare (Directive 2010/63/ EU). Charles Rivers provided WT C57BL/6 mice.

## Animal infection models

6–10 mice/group were intranasally infected with $5.10^5$ Colony Forming Units (CFUs) of *P. aeruginosa* PP34 strain (*ExoU*[+]) or its isogenic mutant (*ExoU*[−]) and animal survival was followed over 40–50 hours after infection. When specified, mice were intraperitoneally treated with 100μL of PBS or ferrostatin-1 (6mg.k[-1]) 4–6 hours before intranasal infections with bacterial strains.

Regarding bacterial loads assays, 6–10 mice/group were intranasally infected with $2.10^5$ bacteria for 24 hours, and Bronchoalveaolar (BALs), lung spleen and blood bacterial numbers were evaluated using CFU plating. BAL fluids (BALFs) were also used to address cytokine, alarmin and lipid levels using ELISA, EIA and colorimetric kits. There were no randomization or blinding performed.

## Histological experiments and scoring

Mice were intraperitoneally treated with 100μL of PBS or ferrostatin-1 (6mg.k[-1]) 4–6 hours before intranasal infections with sub-lethal doses ($2.10^5$ CFUs) of *exoU*-expressing *P. aeruginosa*. 6 hours later, lung tissues were fixed for 48 h in 10% buffered formalin, washed 3 times in ethanol 70% and embedded in paraffin. 5 μm sections were stained with hematoxylin and eosin (HE). Histopathological scoring from 0 to 3 were attributed based on the severity of peribronchial, perivascular, and interstitial cell infiltration, resulting in a maximum score of 9.

## Bacterial cultures

*P. aeruginosa* (PP34, PA103, CHA, PAO1, PA14) bacteria and their isogenic mutants were grown overnight in Luria Broth (LB) medium at 37˚C with aeration and constant agitation in the presence or absence of EGTA (10mM) to ensure T3SS expression. Bacteria were sub-cultured the next day by dilution overnight culture 1/50 and grew until reaching an optical density (OD) O.D600 of 0.6–1. Bacterial strains and their mutants are listed in Table 1.

## Bone Marrow-derived Macrophage (BMDMs), Eosinophil (BMDEs) or Neutrophil (BMDNs) isolation and culture

Murine Bone Marrow-Derived Macrophages (BMDMs) from bone marrow progenitors were differentiated in DMEM (Invitrogen) supplemented with 10% v/v FCS (Thermo Fisher Scientific), 10% v/v MCSF (L929 cell supernatant), 10 mM HEPES (Invitrogen), and nonessential amino acids (Invitrogen) for 7 days as previously described [85].

Murine Bone Marrow-Derived Eosinophils were differentiated *in-vitro* from bone marrow as previously described [86]. cells were resuspendent and cultured at $10^6$/mL in RPMI glutamax medium with HEPES containing 20% FBS, 100 IU/ml penicillin and 10 μg/ml streptomycin, 1 mM sodium pyruvate (Life Technologies), and 50 μM 2-ME (Sigma-Aldrich) supplemented with 100 ng/ml stem cell factor (SCF; PeproTech) and 100 ng/ml FLT3 ligand (FLT3-L; PeproTech) from days 0 to 4. On day 4, the medium containing SCF and FLT3-L was replaced with medium containing 10 ng/ml recombinant mouse (rm) IL-5 (R&D Systems) only. Medium was replaced every 4 days and the concentration of the cells was adjusted each time to106/ml. After 10 to 14 days of culture, cells were recovered by gentle pipetting and used as Eosinophils in our experiments. Over 95% of cells had the standard phenotype of Eosinophils: CD11b+ Siglec F + after FACS analysis.

**Table 1. Resource of reagents used in this study.** Information and reagents are available upon request to Etienne.meunier@ipbs.fr.

| REAGENT or RESOURCE | SOURCE | IDENTIFIER |
|---|---|---|
| Antibodies | | |
| GPX4, 1/1000 | abcam | ab125066 |
| ExoU, 1/1000 | Ina Attree/CNRS, France. | [37] |
| CYPOR 1/1000 | abcam | ab180597 |
| Gapdh 1/10000 | Gentex | GTX100118 |
| Goat anti-Rabbit HRP (1/10000) | Advansta | R-05072-500 |
| Bacterial and Virus Strains | | |
| PAO1 | J. Buyck/Univ of Poitiers/France | N.A. |
| PP34 | Ina Attree/CNRS, France. | [37] |
| PP34*exoU*⁻ | Ina Attree/CNRS, France. | [37] |
| PP34*exoU*^S142A | Ina Attree/CNRS, France. | [37] |
| CHA | Ina Attree/CNRS, France. | [37] |
| CHAdST | Ina Attree/CNRS, France. | [37] |
| CHAdST*exoU*⁺ | Ina Attree/CNRS, France. | [37] |
| PA103 | J. Buyck/Univ of Poitiers/France | N.A. |
| PA103*exoU*⁻ | J. Buyck/Univ of Poitiers/France | N.A. |
| PA14 | J. Buyck/Univ of Poitiers/France | N.A. |
| PA14 *exoU*⁻ | J. Buyck/Univ of Poitiers/France | N.A. |
| Biological Samples | | |
| Human lung biopsy | Hospital of Toulouse | CHU 19 244 C CNRS 205782 |
| Human blood | EFS | 21PLER2017-0035AV02 |
| Chemicals, Peptides, and Recombinant Proteins | | |
| Recombinant ExoU | This study | [31] |
| Recombinant ExoUS142A | This study | [31] |
| FCS | Fisher Scientific | 16010–159 |
| mMCSF | L929 cell supernatant | NA |
| HEPES | Fisher Scientific | SH30237.01 |
| Non-essential amino acids | Invitrogen | |
| ECL Clarity Max Substrate | BioRad | 1705060 |
| ECL Clarity Max Substrate | BioRad | 1705062 |
| Western Blot Strip Buffer | Diagomics | R-03722-D50 |
| Tris base | euromedex | 200923-A |
| SDS ultra-pure (4x) | Euromedex | 1012 |
| Acrylamide / Bisacrylamide 37.5/1 30% | Euromedex | EU0088-B |
| Temed | Sigma | T9281-25ML |
| Ammonium persulfate | Sigma | 248614-100g |
| Page Ruler 10–180 kDa | Fisher Scientific | 15744052 |
| Triton X-100 | Euromedex | 2000 |
| DMEM | Fisher Scientific | 41965–039 |
| LB | Fisher Scientific | BP1426-2 |
| LB Agar | INVITROGEN | 22700025 |
| Roche protease inhibitor cocktail | Sigma | 000000011697498001 |
| BSA | SIGMA | A9647-100G |
| Propidium iodide | Invitrogen | P3566 |
| Beads Neutrophils human | Miltenyi biotec | 130-104-434 |
| Beads Neutrophils murine | Miltenyi biotec | 130-120-337 |
| Kit de coloration bleue et fixable des cellules mortes LIVE/DEAD pour excitation UV | ThermoFisher Scientifique | L34961 |

*(Continued)*

**Table 1.** (Continued)

| REAGENT or RESOURCE | SOURCE | IDENTIFIER |
|---|---|---|
| APC/Cyanine7 anti-mouse CD45 Antibody | BioLegend | 103116 |
| PE/Dazzle 594 anti-human CD64 Antibody | BioLegend | 305032 |
| FITC anti-mouse MERTK (Mer) Antibody | BioLegend | 151504 |
| CD170 (Siglec F) Monoclonal Antibody (1RNM44N), Super Bright 780, | eBioscience | 78-1702-82 |
| Ly-6G Monoclonal Antibody (1A8-Ly6g), APC | eBioscience | 17-9668-82 |
| Brilliant Violet 650 anti-mouse/human CD11b Antibody | BioLegend | 101259 |
| Brilliant Violet 421 anti-mouse Ly-6C Antibody | BioLegend | 128032 |
| PE/Cyanine7 anti-mouse CD11c Antibody | BioLegend | 117318 |
| Eosinophil differentiation cocktail (IL-5) | R&D Systems | 405-ML-005 |
| Eosinophil differentiation cocktail (SCF) | Biolegend | 579706 |
| Eosinophil differentiation cocktail (Flt-3) | Biolegend | 550706 |
| Puromycin | ThermoFisher Scientifique | A1113803 |
| G418 (Geneticin) | invivoGen | ant-gn-1 |
| Blasticidin | nvivoGen | ant-bl-1 |
| Cumene hydroperoxide | Sigma-Aldrich | 247502-5G |
| RSL3 | Sigma-Aldrich | SML2234 |
| Ferrostatin-1 | Sigma-Aldrich | SML0583 |
| Liproxstatin-1 | Sigma-Aldrich | SML1414 |
| DFO | Sigma-Aldrich | D9533 |
| a-tocopherol | Sigma-Aldrich | 258024 |
| MAFP | Sigma-Aldrich | M2689 |
| PPOH | CaymanChem | 75770 |
| Cox1 inhibitor | Ab142904 (Abcam) | Ab142904 |
| Cox2 inhibitor | NS 398 (Abcam) | Ab120295 |
| cPLA2 assay kit | Cayman Chemical | 765021 |
| CD14+ beads | Miltenyi biotec | 130-050-201 |
| RPMI | Fisher Scientific | 72400–021 |
| OPTIMEM | Fisher Scientific | 31985–04 |
| Z-VAD | Invivogen | tlrl-vad |
| TPCA-1 | Tocris | 2559 |
| mTNFa | abcam | ab259411 |
| Olaparib | CaymanChem | 10621 |
| Necrostatin-1s | Sigma-Aldrich | N9037 10MG |
| hMCSF | Miltenyi biotec | 170-076-171 |
| Fisher BioReagents Lymphocyte Separation Medium-LSM | Fisher Scientific | BP2663500 |
| ExoU | This study | N.A. |
| ExoUS142A | This study | N.A. |
| **Human bronchial organoid culture reagents** | | |
| Advanced DMEM/F12 | Invitrogen | 12634028 |
| Gibco L-Glutamine (200 mM) | Fisher | 11500626 |
| Hepes 1 M | Fisher | 11560496 |
| Penicillin/Streptomycin | Fisher | 11548876 |
| Primocin | Invivogen | ant-pm-1 |
| Basic Media | In house | NA |
| RspoI | In house | NA |
| Noggin | In house | NA |
| B27 | Gibco/Invitrogen | 17504044 |

(Continued)

**Table 1.** (Continued)

| REAGENT or RESOURCE | SOURCE | IDENTIFIER |
|---|---|---|
| N-Acetylcysteine | Sigma | A9165-5g |
| Nicotinamide | Sigma | N0636 |
| Y-27632 | Cayman | 10005583 |
| A83-01 | Tocris | 2939 |
| SB 202190 | Sigma | S7067 |
| FGF-7 | Peprotech | 100–19 |
| FGF-10 | Peprotech | 100–26 |
| **Critical Commercial Assays** | | |
| mIL-1alpha ELISA kit | Fisher Scientific | 88-5019-88 |
| mIL-36g ELISA kit | Ray Biotech | ELM-IL36G |
| LDH Cytotoxicity Detection Kit | Takara | MK401 |
| mTNFalpha ELISA kit | Fisher Scientific | 88-7324-22 |
| mIL-33 ELISA kit | Fisher Scientific | 88-7333-88 |
| mHMGB1 ELISA kit | Clinisciences | LS-F4040-1 |
| TBAR MDA colorimetric kit | Cayman | 10009055 |
| PGE2 EIA Kit | Cayman | 514010 |
| LTB4 EIA kit | Cayman | 520111 |
| 8-PGF2 EIA kit | Cayman | 516351 |
| H2DCFDA ROS detecting probe | Invitrogen | D399 |
| C11 bodipy phospholipid peroxide detection probe | Invitrogen | D3861 |
| ER-Tracker Blue-White DPX, for live-cell imaging | Invitrogen | E12353 |
| **Experimental Models: Cell Lines** | | |
| WT Mouse Bone marrow derived macrophages | This study | |
| Alox5-/- Mouse Bone marrow derived macrophages | This study | |
| Alox12/15-/- Mouse Bone marrow derived macrophages | This study | |
| Nlrc4-/- Mouse Bone marrow derived macrophages | This study | |
| Casp1-/-/Casp11-/- Mouse Bone marrow derived macrophages | This study | |
| GsdmD-/- Mouse Bone marrow derived macrophages | This study | |
| WT Mouse bone marrow derived eosinophils | This study | |
| WT Mouse bone marrow derived neutrophils | This study | |
| Human blood monocyte derived macrophages | This study | |
| Human blood neutrophils | This study | |
| Immortalized WT murine bone marrow derived macrophages | This study | |
| Immortalized Gpx4-/- murine bone marrow derived macrophages | This study | |
| Human Bronchial epithelial cells | This study | |
| Human Alveolar epithelial A549 cell line | This study | |
| Human intestinal epithelial HELA cell line | This study | |
| Experimental Models: Organisms/Strains | | |
| WT C57Bl6J mice | C. Rivers | |
| WT C57Bl6N mice | C. Rivers | |
| Alox5-/- C57Bl6 mice | A.Coste | [79] |
| Alox12/15-/- C57Bl6 mice | A.Coste | [79] |
| Nlrc4-/- C57Bl6 mice | C.Bryant | [80] |
| Casp1-/-/Casp11-/- C57Bl6 mice | B.Py/ Junying Yuan | [81] |
| GsdmD-/- C57Bl6 mice | P.Broz | [82] |
| Human Bronchial organoids | This sudy | [83,84] |
| Oligonucleotides | | |

(Continued)

**Table 1.** (Continued)

| REAGENT or RESOURCE | SOURCE | IDENTIFIER |
|---|---|---|
| Guide Crispr mGpx4- Exon1 Forward | Sigma-Aldrich | GGACGCTGCAGACAGCGCGG |
| Guide Crispr mCypor- Exon1 Forward | Sigma-Aldrich | |
| **Recombinant DNA** | | |
| Plasmid: ExoU | [31] | [31] |
| Plasmid: ExoUS142A | [31] | [31] |
| Plasmid: ExoU^BtU | [31] | [31] |
| LentiGuide-Puro | Feng Zhang lab | Addgene #52963 |
| Lenti-multi-Guide | From Qin Yan | Addgene #85401 |
| pMD.2G | Didier Trono lab | Addgene #12259 |
| p8.91 | Didier Trono lab | N.A. |
| LentiCas9-Blast | Feng Zhang lab | Addgene #52962 |
| **Software and Algorithms** | | |
| Graph Pad Prism 8.0 | | |
| Image J | | |
| Snapgene | GSL Biotech LLC, Chicago, U.S.A | |
| FlowJO | FlowJo LLC | |
| Benchling Software | | |
| Other | | |

Murine Bone Marrow-derived Neutrophils were isolated and purified from fresh bone marrows using Anti-Ly-6G micro bead kit (Miltenyi Biotec). Analysis of cell purity by FACS show that over 95% of cells had the standard phenotype of Neutrophils Ly6G+/Ly6C+.

$2.5x10^5$ BMDMs or $1.10^6$ BMDEs/BMDNs were seeded in 24 well-plates and infected or exposed to various treatments. Regarding ferroptosis experiments, BMDMs were infected with various bacterial strains of *P. aeruginosa* expressing or not *exoU* at an MOI 0.1–1 for various times. When specified, recombinant microbial phospholipases (10ng-1μg) were transfected in BMDMs using Fugene (3μl per 1μg of transfected protein) for 2–4 hours. Compound-induced ferroptosis was achieved using RSL-3 (10μM, 8H) or Cumene hydroperoxide (CuOOH, 500μM, 3H).

When required, BMDMs were pretreated for 2 hours with pharmacological inhibitors necrostatin-1s (40μM), Z-VAD (40μM), olaparib (10μM), ferrostatin-1 (1–40μM), MAFP (50μM), liproxstatin (30μM), a-tocopherol (20μM).

For all stimulations, cell culture medium was replaced by serum-free and antibiotic-free Opti-MEM medium and triggers were added to the cells for various times.

## Cell line culture

Immortalized murine bone-marrow derived macrophages have been described previously [85]. U937 cells were cultured in RPMI glutamax medium containing 10% FBS, 100 IU/ml penicillin and 10 μg/ml streptomycin, 1 mM sodium pyruvate (Life Technologies), and 50 μM 2-ME (Sigma-Aldrich). Medium was renewed every 3 days and the concentration of the cells was adjusted each time to $5x10^5$/ml. A549, HeLa and HBE cells were cultured in DMEM glutamax medium with HEPES containing 10% FBS, 100 IU/ml penicillin and 10 μg/ml streptomycin, 1 mM sodium pyruvate (Life Technologies). When the cells reach approximately 90% confluency, cells are detached with Trypsin 0.05% (Gibco), cell suspension is diluted 1/10 in fresh medium, and placed back in the incubator for culture.

## Purification and generation of human blood neutrophils and monocyte-derived Macrophages

Monocytes were isolated from Peripheral Blood Mononuclear Cells (PBMCs) from the buffy coat of healthy donors obtained from the EFS Toulouse Purpan (France) as described previously [87]. Briefly, PBMCs were isolated by centrifugation using standard Ficoll-Paque density (GE Healthcare) [85]. The blood was diluted 1:1 in phosphate-buffered saline (PBS) pre-warmed to 37°C and carefully layered over the Ficoll-Paque gradient. The tubes were centrifuged for 25 min at 2000 rpm, at 20°C. The cell interface layer was harvested carefully, and the cells were washed twice in PBS (for 10 min at 1200 rpm followed by 10 min at 800 rpm) and re-suspended in RPMI-1640 supplemented with 10% of foetal calf serum (FCS), 1% penicillin (100 IU/mL) and streptomycin (100 μg/ml). Monocytes were separated from lymphocytes by positive selection using CD14+ isolation kit (Myltenyi biotec). To allow differentiation into monocyte-derived macrophages, cells were cultured in RPMI medium (GIBCO) supplemented with 10% FCS (Invitrogen), 100 IU/ml penicillin, 100μg/ml streptomycin, 10 ng/ml M-CSF for 7 days.

Human blood neutrophils were isolated from whole blood of healthy donors obtained from the EFS Toulouse Purpan (France). Neutrophils were enriched using MACSxpress Whole Blood Neutrophil Isolation Kit whole blood neutrophil isolation kit (Myltenyi biotec) according to manufacturer instructions. Red blood cells (RBC) were removed by 10 min incubation in RBC Lysis Buffer (BioLegend).

## Genetic invalidation of *Gpx4 and Cypor* genes in immortalized BMDMs

Targeted genes were knocked-out using the crispr/cas9 system in immortalized BMDMs. Single guide RNAs (sgRNA) specifically targeting *Gpx4* exon1 (for 5' GGACGCTGCAGACAGCGCGG 3' *Cypor* exon2 (for 5' AGTGTCTCTATTCAGCACAA 3' were designed using Benchling tool (Benchling.com), and oligonucleotides were synthesized by Sigma-Aldrich. Crispr guide RNA oligonucleotides were hybridized and subsequently cloned into the vector Lenti-gRNA-Puromycin using BsmBI restriction enzyme (Addgene 52963, Feng Zhang lab). Generated constructs were then transfected in lipofectamine 2000 into HEK293T for 48 hours together with the lentiviral packaging vector p8.91 (Didier Trono lab, EPFL, Switzerland) and the envelop coding VSVg plasmid (pMD.2G, Addgene 12259, Didier Trono lab). Viral supernatants were harvested, filtered on 0.45 μm filter and used to infect cells expressing Cas9 (1,000,000 cells/well in 6-well plates. Efficient infection viral particles was ensured by centrifugating cells for 2 h at 2900 rpm at 32°C in presence of 8μg/ml polybrene. 48 h later, medium was replaced and Puromycin selection (10μg/mL) was applied to select positive clones for two weeks. Puromycin-resistant cells were sorted at the single cell level by FACS (Aria cell sorter). Individual clones were subjected to western blotting to confirm the absence of targeted proteins.

## Human bronchial organoid production and maintenance

Airway organoids were derived from lung biopsies as described [83,84]. Briefly, Human lung tissue was provided by the CHU of Toulouse under the CNRS approved protocols CHU 19 244 C and CNRS 205782. All patients participating in this study consented to scientific use of their material. Biopsies (1 mm3) of normal lung tissue adjacent to the tumor obtained from patients who underwent lung resection due to Non-small cell lung carcinoma (NSCLC) were minced and digested with 2 mg ml−1 collagenase (Sigma) on an orbital shaker at 37°C for 1h. The digested tissue suspension was sheared using flamed glass Pasteur pipettes and strained

over a 100-µm cell strainer (Falcon). The resultant single cell suspensions were embedded in 10 mg ml−1 of Cultrex growth factor reduced BME type 2 (R & D Systems) and 40µl drops were seeded on Nunclon Delta surface 24-well plates (Thermo Scientific). Following polymerization, 500 µl of Advanced DMEM/F12 (Invitrogen) supplemented with 1x L-Glutamine (Fisher Scientific), 10mM Hepes (Fisher Scientific), 100 U ml-1 / 100 µg ml-1 Penicillin / Streptomycin (Fisher Scientific), 50 µg ml-1 Primocin (InvivoGen), 10% Noggin (homemade), 10% RspoI (homemade), 1x B27 (Gibco), 1.25mM N-Acetylcysteine (Sigma-Aldrich), 10mM Nicotinamide (Sigma-Aldrich), 5µM Y-27632 (Cayman Chemical), 500nM A83-01 (Tocris Bioscience), 1µM SB 202190 (Sigma-Aldrich), 25 ng ml−1 FGF-7 (PeproTech), 100 ng ml−1 FGF-10 (PeproTech) was added to each well and plates transferred to humidified incubator at 37˚C with 5% CO2. The organoids were passaged every 4 weeks.

## Organoid infections

Before infection, 35µl drops of Matrigel (Fisher Scientific) containing organoids were seeded on Nunclon Delta surface 35x10mm Dish (Thermo Scientific) and 2ml of Advanced DMEM/F12 supplemented with 1x L-Glutamine and 10mM Hepes was added to each plate. Depending on the indicated conditions, organoids were pretreated or no with 40µM Ferrostatin-1 for 1hr before infection. Ferrostatin-1 was maintained throughout the experiment. PP34 *exoU* or *exoU*$^{S142A}$ were grown as previously described until reach OD600 = 1. Bacterial density was adjusted to OD600 = 0.0005, and phenol red added at 0.005% to visualize successful microinjection (2). Injected organoids were individually collected and re-seeded into fresh matrix for subsequent analysis. For time-lapse imaging, injected and stimulated organoids were stained with 50 µg ml-1 Propidium Iodide (Thermo Scientific). Images were acquired every 15 minutes for the duration of experiments under an EVOS M7000 (Thermo Scientific) Imaging System (10x, at 37˚C with 5% CO2). Data was analyzed using Fiji/ImageJ.

## Cell necrosis, alarmin/cytokine and lipid release assays

LDH Cytotoxicity Detection Kit (Takara) was used to determine the percentage of cell lysis. Normalization of spontaneous lysis was calculated as follows: (LDH infected–LDH uninfected)/(LDH total lysis–LDH uninfected)*100.

Murine Il-1α, IL-33, IL-36α, IL-36γ, HMGB1, TNFα, cytokine levels in cell supernatants or in BALFs were measured by ELISA listed in resource Table 1.

Oxidized lipids isoprostanes, eicosanoids PGE2 and LTB4 were detected in cellular supernatants or BALFs using EIA kits listed in resource Table 1.

## Plasma membrane permeabilization assays

Cells are plated at density of 1 x $10^5$ per well in 96-well Plates or at $2x10^5$/well in 24-well plates (Corning 356640) in complete culture medium. The following day, medium is replaced by Opti-MEM supplemented with Propidium iodide (100 ng/ml) or SYTOX green (100ng/mL). Pharmacological inhibitors are added 1h before infection. Red (Propidium Iodide) or green (SYTOX) fluorescence are measured in real-time using Clariostar plate reader or an EVOS7000 microscope, both equipped with a 37˚C cell incubator.

## Malondialdehyde (MDA) assays

Malondialdehyde production was addressed using the MDA lipid peroxidation kit according to the manufacturer's instructions (Abcam, ab118970). Cells were lysed using 500 µl of lysis buffer supplemented with butylated hydroxytoulene. Cell lysates were centrifuged for 10 min

at 13,000 g (RCF) and the supernatants were used for MDA assay. TBA solution was added to each replicate, and samples were then incubated at 95˚C for 1 hour. 100μL of each sample was then processed for fluorometric assay at Ex/Em = 532/553 nm. BAL levels of MDA were normalized to the total protein concentration.

### Recombinant protein production

Plasmids coding for *exoU*[BtU], *exoU* or *exoU*[S142A] were a kind gift from Dara W. Frank's lab. All recombinant proteins were expressed in BL21(DE3) pLysS strain in LB medium, according to Anderson DM et al. [31]. Proteins fused with an N-terminus hexahistidine-tag were purified as previously described with slight modifications. Briefly, after cell harvest, bacteria were lysed by sonication under ice and recombinant proteins were purified by nickel metal affinity chromatography (Takara). After sample concentration, Superose 6 was exchanged for a Superdex 200 size exclusion column (GE Healthcare) as a final purification step. Samples were either used fresh or keep at -80˚C for long-term storage. ExoU and ExoUS142A activities were validated on cellular lysates (**Fig 3I**) based on the advices and experience of our collaborator [37].

### Cytometry quantification of immune cells in mice BAL fluids (BALFs)

C57BL/6 mice received an injection of Ferrostatine (6mg/kg) or PBS as control intraperitoneally. 4-6h after, mice were infected by intranasal instillation of 50 μL of PBS containing or not $5x10^6$ bacteria (PP34) in presence or absence of Ferrostatin-1 (6mg /kg). 18h after infection, BALFs were collected and quality/quantity of immune cells content was assayed by flow cytometry. Briefly, cells were pelleted (1000 rpm, 5 minutes), Red blood cells (RBC) were removed by 10 min incubation in RBC Lysis Buffer (BioLegend), monocytes, macrophages, neutrophils, and eosinophils were subsequently stained with a cocktail of fluorochrome-conjugated antibodies detailed in the "Material and Method" section. Cells were then fixed in 4% PFA before fluorescence associated cell sorting (FACS) analysis using a LSRII instrument. AccuCheck Counting Beads (ThermoFisher) were used to determine absolute cell number. Data analysis and processing were performed using FlowJO software.

### Lipid peroxidation or ROS production

To measure lipid peroxidation or ROS production, cells were first washed with PBS 1X, and then incubated with either C11-BODIPY(581/591) (ThermoFisher) at 2 μM, or H2DCFDA (ThermoFisher) at 10 μM in Opti-MEM medium for 30 min at 37˚C. After three washes with PBS 1X cells are resuspended in Opti-MEM medium and infected/treated in presence or absence of pharmacological inhibitors. After 1-3h of infection, cells are washed with PBS, detached in MACS buffer (PBS-BSA 0,5%-EDTA 2mM) and samples were acquired within one hour using a flow cytometer (BD FORTESSA LSR II or a FACS Calibur). Data were analysed with FlowJO software (version 10). When specified, adherent cells loaded with Bodipy probes where infected at indicated MOIs of *P. aeruginosa* and lipid peroxidation is observed using an EVOS7000 microscope. For live imaging, the GFP brightness threshold was kept equal for all the independent experiments. Mean fluorescence intensity (MFI) was analyzed using Fiji/ImageJ.

### Immunoblotting

Cell lysate generation has been described previously [85]. Briefly, proteins were loaded in 12% SDS-PAGE gels and then transferred on PVDF membranes. After saturation for 1 hour in Tris-buffered saline (TBS) supplemented with 0.05% Tween 20 containing 5% non-fat milk (pH8), membranes were exposed with antibodies at 4˚C overnight (Table 1). Next day,

membranes were washed 3 times in TBS 0.1% Tween 20 and incubated with the corresponding secondary antibodies conjugated to horseradish peroxidase (HRP) (Table 1) for 1h at room temperature. Immunoblottings were revealed using a chemiluminescent substrate ECL substrate (Biorad) and images were acquired on a ChemiDoc Imaging System (Biorad). All antibodies and their working concentrations are listed in **Table 1**.

## (Redox) lipidomic

1 million bone-marrow-derived macrophages were seeded into 6-well plates. Next day, BMDMs were transfected with recombinant ExoU or ExoU$^{S142A}$ proteins (500ng/well) for one hour. Then, supernatant was removed, cells were washed two times in PBS. Finally, 500μL of a cold solution of 50% PBS/50% Methanol was added to cells and samples were transferred to -80˚C for storage and subsequent analyses.

After thawing, lipids were extracted using a methyl-tert-butyl ether (MTBE)-based liquid-liquid extraction method. Cell suspensions (500 μL in PBS/methanol 1:1, v/v) were thawed on ice before adding 100 μL methanol MeOH containing 50 ng each of the internal standards PC (15:0/18:1-d7), PE(15:0/18:1- d7), PG(15:0/18:1-d7), PI(15:0/18:1-d7) and PS(15:0/18:1-d7) (EquiSPLASH, Avanti Polar Lipids). Samples were then transferred into 8-mL screw-cap tubes, and then 1.125 methanol and 5 mL MTBE were added. After vigorous mixing, samples were incubated at room temperature on a tabletop shaker for 45 min. For phase separation, 1.25 mL water was added, and samples were vortexed and centrifuged for 15 min at 2000 x g. The upper organic phase of each sample was carefully removed using a Pasteur pipette, transferred into an empty glass round-bottom tube, and dried under vacuum in a SpeedVac concentrator. The dried lipid extracts were resuspended in 200 μL HPLC mobile phase A/mobile phase B 3:1 (v/v) for targeted lipidomic analysis of oxidized phospholipids. For LC-MS/MS, using a Sciex ExionLC Integrated System, 20 μL of each lipid extract was injected using Column Kinetex 2.6 μm HILIC 100 Å 100x2.1 mm, Phenomenex and a Flow Rate of 200 μL/min. Then, the analyte-specific m/z transition profile was determined and the area under the peak (ion intensity vs. elution time) was calculated using MultiQuant, Sciex software.

Data calculation was performed by doing ratio between the values of "area ratio analyte/ internal standard" of each oxidized phospholipid and its non-oxidized phospholipid. The fold induction in oxidized phospholipid was then calculated by doing a ratio between each oxidized ratio and the non-stimulated condition. Accordingly, the unstimulated condition oxidized ratios were 1 or 0 when no peroxidation was detected in any condition.

## Phospholipase activity measurement

Evaluation of ExoU phospholipase activity was performed using the Cayman Chemical cPLA2 kit and performed as previously described with minor modifications [37]. Briefly, 10 μL of a 1mg/mL (160pmols) solution of recombinant ExoU or ExoU$^{S142}$ proteins were mixed in 96-well plates with 10μL of lysed cell samples and 10μL of Assay Buffer. Then, samples were incubated for 1 hour at room temperature with 250μL of substrate solution (1.5 mM arachidonyl thiophosphatidylcholonie) and then for additional 4 or 16 hours in dark. Reaction was stopped using 25mM solution of DTNB according to manufacturer instructions and absorbance was detected at 405nm using a Clariostar plate reader. Phospholipase activity of ExoU or ExoU$^{S142}$ was calculated as the hydrolysis rate accordingly to the manufacturer instructions.

## Statistical analysis

Statistical data analysis was performed using Prism 8.0a (GraphPad Software, Inc.). We used t-test with Bonferroni correction for comparison of two groups. Data are reported as mean with

SEM. Regarding animal experiments, we used Mann-Whitney tests and mouse survival analysis were done using log-rank Cox-Mantel test. P values in figures have the following meaning; NS non-significant and Significance is specified as $^*p \leq 0.05$; $^{**}p \leq 0.01$, $^{***}p \leq 0.001$.

## Supporting information

**S1 Data. Original immunoblotting membranes.**
(TIF)

**S2 Data. Numerical values obtained in the current study.**
(XLSX)

**S1 Fig. ExoU-dependent lung pathology in mice occurs in an inflammasome-independent manner. (A)** Survival of WT, *Casp1*$^{-/-}$/*Casp11*$^{-/-}$, *Nlrc4*$^{-/-}$ and *GsdmD*$^{-/-}$ mice intranasally infected (n = 6 animals per condition) with $5.10^5$ CFUs of *P. aeruginosa* PP34. Graphs represent one experiment (6 mice/group) out of three independent *in vivo* experiments. NS: Not significant using Log-rank Cox-Mantel test for survival comparisons. **(B)** Bronchoalveolar (BAL) and lung bacterial loads from WT, *Casp1*$^{-/-}$/*Casp11*$^{-/-}$, *Nlrc4*$^{-/-}$ and *GsdmD*$^{-/-}$ mice (n = 6) 18 hours after intranasal infection with $5.10^5$ CFUs of *P. aeruginosa* PP34. Graphs represent one experiment (6 mice/group) out of three independent *in vivo* experiments. NS: Not significant using Mann-Whitney analysis test.
(TIF)

**S2 Fig. Lipid peroxidation contributes to ExoU-induced necrosis in various cell types. (A, B)** Measure of LDH release in various human and murine cell types infected with various *P. aeruginosa* strains expressing or not *exoU* in presence of Ferrostatin-1 (Fe1, 10μM) for 2 hours. **(C)** LDH release in BMDMs transfected with recombinant ExoU (100ng) or its catalytically inactive mutant ExoU$^{S142A}$, in presence of MAFP (50μM) or Ferrostatin-1 (Fe1, 10μM) for 3 hours. $^{***}p \leq 0.001$, T-test with Bonferroni correction. **(D)** Immunoblotting of ExoU secretion by *P. aeruginosa* in presence of ferrostatin-1 (20μM). Star (*) show non-specific bands. **(E)** Measure of bacterial growth (O.D 600) in presence or absence of ferrostatin-1 (10, 20μM) for 14 hours). **(F)** Measure of LDH release in *Nlrc4*$^{-/-}$ BMDMs infected with PP34 (MOI5) in presence of Ferrostatin-1 (Fe1, 10μM) for 3 hours. Each hour, fresh Ferrostatin-1 (10μM) was added to cells (+) or not (φ). "pi" refers to post-infection.
(TIF)

**S3 Fig. Lipid peroxidation fuels ExoU-dependent necrosis. (A)** ROS production in WT BMDMs transfected with ExoU or its catalytically dead mutant ExoU$^{S142A}$ for 45 minutes using H2DCFDA (1μM) probe. **(B)** Cytometry detection and quantification of (phospho)lipid peroxidation using the probe C11-bodipy in WT BMDMs infected with PP34$^{ExoU+}$ or PP34$^{ExoU-}$ (MOI 5) for 1 hour. Sample were acquired using FACSCalibur (BD). The graph shows the mean+/-SEM of one experiment performed in triplicate out of three independent experiments. $^*P \leq 0.05$, for the indicated comparisons using t-test with Bonferroni correction. **(C)** Lipidomic analysis of the relative amount of each phospholipid upon rExoU transfection analysed in **Fig 3B**. **(D)** Representative microscopy images and time course experiment of propidium iodide uptake in WT BMDMs transfected with rExoU or its catalytically inactive mutant ExoU$^{S142A}$ (500ng) in presence or not of ferrostatin-1 (Fe1, 10μM). Images show two independent experiments, each performed three times at 45 minutes or 3 hours post transfection. **(E)** Immunoblotting of Crispr Cas9-mediated *Cypor* gene deletion in immortalized (i)BMDMs or of *Cypor*-deficient HELA cells. The *Cypor*#2 (red) was selected for further analysis. GFP means that cells were transduced with sgRNA targeting *Gfp* and used as control. **(F)** Cytometry detection and

quantification of (phospho)lipid peroxidation using the probe C11-bodipy in WT or *Cypor*[-/-] immortalized (i)BMDMs pre-treated or not for 1 hour with CuOOH (20μM) in presence or absence of Ferrostatin-1 (20μM) and then infected with PP34[ExoU+] or PP34[ExoU-] (MOI 5) for 1 hour. Sample were acquired using FACSCalibur (BD). The graphs shows the mean+/-SEM of one experiment performed in triplicate out of three independent experiments. *$P \leq 0.05$, **$P \leq 0.001$, for the indicated comparisons using t-test with Bonferroni correction. **(G)** Immunoblotting of Crispr Cas9-mediated *Gpx4* gene deletion in immortalized BMDMs. The Gpx4#1 (red) was selected for further analysis. CD8 and GFP means that cells were transduced with sgRNA targeting *Gfp* or *Cd8* genes and used as controls. **(H)** Cytometry detection and quantification of phospholipid peroxidation using the probe C11-bodipy in immortalized WT or *Gpx4*[-/-] BMDMs using a fortessa cytometer. **(I)** Measure of 8-iso PGF2α isprostane in cell supernatant in WT BMDMs pre-treated or not for 1 hour with CuOOH (20μM) in presence or absence of Ferrostatin-1 (20μM) and then transfected with rExoU (500ng) for 3 hours. ***$p \leq 0.001$, T-test with Bonferroni correction. **(J)** PGE2 and LTB4 eicosanoid release in WT BMDMs pre-treated or not for 1 hour with CuOOH (20μM) and then transfected with 100ng of ExoU or its catalytically dead mutant ExoU[S142A] for 3 hours.
(TIF)

**S4 Fig. Ferrostatin-1 protects mice against ExoU-induced lung pathology. (A)** Gating strategy to analyse Immune cell populations in bronchoalveolar fluids (BALFs). Immune cells were identified as CD45+ cells. Among CD45+ cells, different subset of immune cells including Interstitial/Alveolar Macrophages, Eosinophils and Neutrophils are identified based on specific cell surface marker expression.
(TIF)

**S1 Graphical Abstract. Host lipid peroxidation fuels ExoU-induced cell necrosis-dependent pathology.** In resting cells or in cells with induced lipid peroxidation (e.g. ferroptosis pathway), ExoU (purple) becomes hyper-activated by host cell peroxidised phospholipids, which drives an exacerbated cell necrosis, alarmin and lipid release and contributes to the subsequent pathology. Consequently, targeting lipid peroxidation (ferrostatin-1) inhibits ExoU-dependent cell necrosis and attenuates the host deleterious consequences. EM and SB used Biorender.com to create this figure.
(TIF)

**S1 Movie. Live cell imaging of uninfected immortalized murine *Nlrc4*[-/-] BMDMs cell death using SYTOX green.** 1 "time point" corresponds to 150s.
(AVI)

**S2 Movie. Live cell imaging of uninfected immortalized murine *Nlrc4*[-/-] BMDMs cell death in presence of 20μM of ferrostatin-1 using SYTOX green.** 1 "time point" corresponds to 150s.
(AVI)

**S3 Movie. Live cell imaging of immortalized murine *Nlrc4*[-/-] BMDMs cell death infected with *exoU*-expressing *P. aeruginosa* (MOI1) using SYTOX green.** 1 "time point" corresponds to 150s.
(AVI)

**S4 Movie. Live cell imaging of immortalized murine *Nlrc4*[-/-] BMDMs cell death infected with *exoU*-expressing *P. aeruginosa* (MOI1) in presence of ferrostatin-1 (20μM) using SYTOX green.** 1 "time point" corresponds to 150s.
(AVI)

**S5 Movie. Live cell imaging of immortalized murine *Nlrc4*<sup>-/-</sup> BMDMs cell death infected with *exoU*-deficient *P. aeruginosa* (MOI1) using SYTOX green.** 1 "time point" corresponds to 150s.
(AVI)

**S6 Movie. Live cell imaging of immortalized murine *Nlrc4*<sup>-/-</sup> BMDMs cell death infected with *exoU*-deficient *P. aeruginosa* (MOI1) in presence of ferrostatin-1 (20µM) using SYTOX green.** 1 "time point" corresponds to 150s.
(AVI)

**S7 Movie. Live cell imaging of uninfected human bronchial organoids using Propidium Iodide up to 12 hours.**
(AVI)

**S8 Movie. Live cell imaging of uninfected human bronchial organoids in presence of ferrostatin-1 (40µM) using Propidium Iodide up to 12 hours.**
(AVI)

**S9 Movie. Live cell imaging of human bronchial organoids microinjected with *exoU*-expressing *P. aeruginosa* using Propidium Iodide up to 12 hours.**
(AVI)

**S10 Movie. Live cell imaging of human bronchial organoids microinjected with *exoU*-expressing *P. aeruginosa* in presence of ferrostatin-1 (40µM) using Propidium Iodide up to 12 hours.**
(AVI)

**S11 Movie. Live cell imaging of human bronchial organoids microinjected with *exoU*-deficient *P. aeruginosa* using Propidium Iodide up to 12 hours.**
(AVI)

**S12 Movie. Live cell imaging of human bronchial organoids microinjected with *exoU*-deficient *P. aeruginosa* in presence of ferrostatin-1 (40µM) using Propidium Iodide up to 12 hours.**
(AVI)

## Acknowledgments

*Alox5*<sup>−/−</sup> and *Alox12/15*<sup>−/−</sup> mice came from Jaxson Laboratory. *Nlrc4*<sup>−/−</sup> mice were provided by Clare E. Bryant and generated by Millenium, *GsdmD*<sup>−/−</sup> mice came from P. Broz (Univ of Lausanne), and *Casp1*<sup>−/−</sup>/ *Casp11*<sup>−/−</sup> came from B. Py (ENS Lyon, France) and were generated by Junying Yuan (Harvard Med School, Boston, USA). Phospholipid redox lipidomic experiments were performed by Cayman Chemical Company (Ann Arbor, USA). Authors also acknowledge the animal facility and Cytometry/microscopy platforms of the IPBS institute. The graphical abstract was generated using Biorender.com.

## Author Contributions

**Conceptualization:** Rémi Planès, Etienne Meunier.

**Data curation:** Salimata Bagayoko, Rémi Planès, Etienne Meunier.

**Formal analysis:** Salimata Bagayoko, Céline Cougoule, Rémi Planès, Etienne Meunier.

**Funding acquisition:** Yoann Rombouts, Céline Cougoule, Rémi Planès, Etienne Meunier.

**Investigation:** Stephen Adonai Leon-Icaza, Miriam Pinilla, Audrey Hessel, Karin Santoni, David Péricat, Pierre-Jean Bordignon, Flavie Moreau, Elif Eren, Aurélien Boyancé, Céline Berrone, Yoann Rombouts, Céline Cougoule.

**Methodology:** Emmanuelle Naser, Nino Iakobachvili, Arnaud Metais.

**Project administration:** Salimata Bagayoko, Rémi Planès, Etienne Meunier.

**Resources:** Lise Lefèvre, Geanncarlo Lugo-Villarino, Agnès Coste, Ina Attrée, Dara W. Frank, Hans Clevers, Peter J. Peters.

**Supervision:** Rémi Planès, Etienne Meunier.

**Validation:** Salimata Bagayoko, Rémi Planès, Etienne Meunier.

**Visualization:** Salimata Bagayoko, Céline Cougoule, Rémi Planès, Etienne Meunier.

**Writing – original draft:** Salimata Bagayoko, Rémi Planès, Etienne Meunier.

**Writing – review & editing:** Salimata Bagayoko, Rémi Planès, Etienne Meunier.

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
