## [Decision Letter · Decision Letter 0]

6 Apr 2021

Dear Dr. Meunier,

Thank you very much for submitting your manuscript "Host phospholipid peroxidation fuels ExoU-dependent cell necrosis and supports Pseudomonas aeruginosa-driven pathology" for consideration at PLOS Pathogens. As with all papers reviewed by the journal, your manuscript was reviewed by members of the editorial board and by several independent reviewers. In light of the reviews (below this email), we would like to invite the resubmission of a significantly-revised version that takes into account the reviewers' comments.

Please pay particular attention to the following reviewer suggestions and give them due consideration.

As requested by Reviewers 1 and 2, clarify how ExoU-mediated necrosis deviates from ferroptosis.As requested by Reviewer 3, please provide additional time course experiments to show the time of the first signs of necrosis. Please also assess the levels of oxidized lipids in cultures infected with viable bacteria (ExoU+ and ExoU-) to strengthen the results obtained with recombinant ExoU.Please revise the graphical abstract to include depictions of the time sequence and of the balanced state as well as hyper- and hypo-lipid peroxidation states.

We cannot make any decision about publication until we have seen the revised manuscript and your response to the reviewers' comments. Your revised manuscript is also likely to be sent to reviewers for further evaluation.

Sincerely,

Gregory P Priebe, M.D.

Guest Editor

PLOS Pathogens

David Skurnik

Section Editor

PLOS Pathogens

Kasturi Haldar

Editor-in-Chief

PLOS Pathogens

orcid.org/0000-0001-5065-158X

Michael Malim

Editor-in-Chief

PLOS Pathogens

orcid.org/0000-0002-7699-2064

Reviewer's Responses to Questions

**Part I - Summary**

Reviewer #1: The research article entitled, “Host phospholipid peroxidation fuels ExoU-dependent cell necrosis and supports Pseudomonas aeruginosa-driven pathology” is an interesting and important study that further clarifies the mechanism by which ExoU causes host cell lysis in a variety of cell types. The authors present strong evidence to support the hypothesis that ExoU injected into the cytosol of host cells by bacterial type III secretion system utilizes the natural process of membrane lipid peroxidation to enhance ExoU phospholipase A2 activity leading to rapid necrosis. Oxidized lipids serve as signals for the cellular process of ferroptosis and increasing or inhibiting ferroptosis through modulation of lipid peroxidation appears to modulate ExoU activity in parallel. Two main conclusions remain somewhat unclear from these studies;

(1) How does exoU mediated necrosis deviate from ferroptosis? If lipid peroxidation enhances both ExoU-mediated necrosis and feroptosis, is there a means to prevent ferroptosis is the context of increased lipid peroxidation that would preserve ExoU-mediated necrosis or vice versa? Are other triggers of necrosis resistant to feostatin-1 treatment when they do not depend on lipid peroxidation? In a balanced lipid peroxidative state is would seem that the presence of ExoU exploits the basal state of lipid peroxidation to cleave at the sn2 position and this heightened PLA2 activity may serve to destabilize the membrane, however, in a heightened lipid peroxidative state where the cell is destined for ferroptosis, does ExoU necrosis override this process and what are distinguishing consequences, if any, between a cell undergoing ExoU-mediated exocytosis or ferroptosis? Is alarmin release an exclusive indicator of cell necrosis or does this occur during ferroptosis? In Figure 2D, is there a reason why there is no phenotype in alox12/15-/- cells? Is 15LOX-1 in mice critical means of oxidizing membrane phospholipids?

(2) For in vivo studies, what is the relative contribution of cellular necrosis versus elaboration of eicosanoid inflammatory mediators and recruitment of immune cells (which precedes necrosis) toward the lung pathology? What cell types in vivo are the most consequential target of ExoU that result in pathology. Since ferrostatin-1 inhibits lipid peroxidation, it would be predicted by the conclusion, “ExoU-targeted peroxidised phospholipids might increase its phospholipase activity toward all phospholipids (peroxidized or not)”, that both cellular necrosis and enhanced release of inflammatory mediator would be interfered with by ferrostatin-1 treatment. The authors draw the conclusion that, “Although a pathological function of recruited immune cells such as neutrophils is probable, we hypothesize that ferrostatin-1-inhibited resident alveolar macrophage death in response to exoU-expressing P. aeruginosa might confer an improved immune protection characterized by lower immune cell recruitment and lower tissue damages”. It could also be hypothesized that ferrostatin-1 reduces the magnitude of eicosanoid generation for mediators such as LTB4, which result in less neutrophil recruitment and less immune cell-mediated damage. Further studies to experimentally disconnect necrosis from enhanced eicosanoid generation by ExoU would be necessary to delineate the relative role in pathology. Such insight would better resolve in vivo mechanisms to inform the development of treatment strategies. These issues warrant being addressed within the Discussion.

The graphic abstract could be significantly improved to better clarify the story of ExoU impact by including time sequence as well as balanced state, hyper and hypo-lipid peroxisation states.

Reviewer #2: Bakgayoko et al., reports a fundamental aspect of host-pathogen interactions, namely the interference with signaling pathways of their host cell by intracellular pathogens. This is a carefully written and experimentally well conducted study that demonstrates the exploitation of host phospholipids by P. aeruginosa. Here the authors demonstrate, a phospholipase from P. aeruginosa, ExoU, triggers host lipid peroxidation and induces ROS mediated necrosis in the host cell. ExoU mediated necrosis is counter effected by Ferrostatin-1 further validates the hypothesis. Overall, this is an interesting study in the context of understanding the P. aeruginosa induced pathology. If the concerns listed below can be addressed, I recommend this manuscript for publication.

Reviewer #3: The manuscript presents a novel point of view about the exploitation of host oxidative mechanisms by Pseudomonas aeruginosa. According to the authors. P. aeruginosa uses the endogenous basal lipid peroxidation by the virulence factor ExoU to promote cell death. The study was well conducted and shows scientific relevance.

**Part II – Major Issues: Key Experiments Required for Acceptance**

Reviewer #1: In aggregate this is a well-designed study with important implications. The biggest weakness lies in effectively articulating the findings. The graphic abstract is insufficient and there are some confusing statements and inconsistencies leaving the reader a little confused despite compelling data. The role of ferroptosis in all this and how it interplays with cellular necrosis is not addressed. Does ExoU interfere with ferroptosis in order to drive necrosis or is ExoU functioning at a basal non-ferroptosis state? If you trigger ferroptosis, thereby enhancing ExoU-driven necrosis, does the ferroptosis process have any consequence to the cell or is necrosis the dominant process in that context?

Reviewer #2: Major points:

1) Explain how did the authors validate successful ExoU mutation?

2) I strongly feel Figure S1 which demonstrates the effect is inflammasome independent is important result in the context of this study and suggest the data should be shown in the main text rather than in supplement.

3) What are the cellular sources of ROS production? And the source of the lipids that are susceptible for ExoU induced peroxidation. Probably, H2DCFDA and Bodipy stained immunofluorescence assay with organelle markers can address this. I understand these experiments are time consuming and if the authors feel it is out of scope of the current study, I suggest discussing couple of lines on this, based on what is known already.

4) When the effect of ExoU is suppressed by Ferrostatin-1 and Liproxstatin-1, why does the authors still refer it as necrosis and rule out the possibility of ferroptosis. Explanation needed.

5) As rExoU failed to induce lipid peroxidation (Fig 3A), I doubt the oxidized lipids could be of bacterial origin which could result in increase in ROS in host cell and collapse. I strongly suggest having an IFA read out for C11-Bodipy to locate the source of lipid peroxidation.

Reviewer #3: To improve the manuscript, I have the following suggestions:

1. The hypothesis is based on the findings presented in figure 3, which show a reduced level in oxidized lipids in cultures transfected with recombinant ExoU in comparison to the observed in non-transfected cultures.

To achieve these results, the authors performed the experiments using a 45 min time-point because at this time they did not detect any plasma membrane permeabilization by propidium iodide uptake (propidium uptake was monitored at 45 min and 3 hours). Therefore, the authors considered that, by using this time-point, they would exclude the involvemet of the ExoU-induced cell necrosis in the decrease of peroxidised phospholipids. However, Since Sato et al., (2003) have reported that ExoU is able to decrease the viability of yeast cells only 30 minutes after ExoU induction from a low copy number plasmid, the authors should show the exact time (between 45 min and 3 h) of the first signs of necrosis in their model.

2. The authors should show the levels of oxidized lipids in cultures infected with viable bacteria (ExoU+ and ExoU-) to strengthen the data obtained with recombinant ExoU.

3. Based on Figure 2E, the authors reported that ferrostatin-1 delayed the ExoU-induced cell necrosis, but did not conclude whether this effect had resulted from the ferrostatin-1 instability over time or from a lipid peroxidation-independent cell death mediated by the phospholipase activity of ExoU. Since ferrostatin1 did not interfere in bacterial viability or ExoU secretion, the authors should add fresh medium containing ferrostatin-1 to solve this question.

**Part III – Minor Issues: Editorial and Data Presentation Modifications**

Reviewer #1: Line 88: a space is missing “balancein”

Line 168: and extra space is present “E xoU”

Line 236: Figure S3C is mentioned before S3B.

Line 262-264 is an incomplete sentence.

Line 325: a space is missing “toevaluate” and the word “rigger” is missing a “t” at the beginning. Furthermore, I recommend starting a new paragraph here as the authors begin describing an entirely different model system.

Line 332: This final sentence of the results is contradictory to earlier claims, “Altogether, our results identified lipid peroxidation as a pathological mechanism induced by the P. aeruginosa ExoU phospholipase both in mice and in human bronchial organoids.”. Earlier, authors make the careful point, concluded through experimentation, that ExoU does not induce lipid peroxidation, but rather exploits existing baseline lipid peroxidation to execute necrosis. This sentence undercuts that message, unless the authors are suggesting that ExoU induces lipid peroxidation in vivo but not in vitro. If that is the case, there does not seem to be specific evidence for this.

Figure S3F is confusing, the labeling says CuOOH but the legend says ferrostatin-1. From the text it seems the error occurred in the legend, as ferrostatin-1 was not involved in this experiment.

Reviewer #2: Minor points:

1) Discuss whether the other patatin family proteins play a role in the absence of ExoU?

2) Fig S2( C) What does the asterisk represents?

3) Does ExoU has a role in regulating transferrin uptake by the host cell?

Reviewer #3: 1. There are some typing mistakes, and the manuscript should be carefully revised.

2. The legend of Figure 2E describes a graphic with the time course measure of plasma membrane permeabilization using propidium iodide incorporation in Nlrc4-/- BMDMs infected with PP34, PP34exoU- or PP34exoUS142A in presence of Ferrostatin-1 (Fe1, 20μM). However, the figure shows only non-infected BMDM as well as PP34- and PP34exoUS142A-infected cells but no data from exoU- infected cells.

PLOS authors have the option to publish the peer review history of their article (what does this mean?). If published, this will include your full peer review and any attached files.

Reviewer #1: No

Reviewer #2: No

Reviewer #3: No
---

## [Decision Letter · Decision Letter 1]

23 Jul 2021

Dear Dr. Meunier,

Thank you very much for submitting your manuscript "Host phospholipid peroxidation fuels ExoU-dependent cell necrosis and supports Pseudomonas aeruginosa-driven pathology" for consideration at PLOS Pathogens. As with all papers reviewed by the journal, your manuscript was reviewed by members of the editorial board and by several independent reviewers. The reviewers appreciated the attention to an important topic. Based on the reviews, we are likely to accept this manuscript for publication, providing that you modify the manuscript according to the review recommendations.

Sincerely,

Gregory P Priebe, M.D.

Guest Editor

PLOS Pathogens

David Skurnik

Section Editor

PLOS Pathogens

Kasturi Haldar

Editor-in-Chief

PLOS Pathogens

orcid.org/0000-0001-5065-158X

Michael Malim

Editor-in-Chief

PLOS Pathogens

orcid.org/0000-0002-7699-2064

Reviewer Comments (if any, and for reference):

Reviewer's Responses to Questions

**Part I - Summary**

Reviewer #1: The revised manuscript entitled, “Host phospholipid peroxidation fuels ExoU-dependent cell necrosis and supports Pseudomonas aeruginosa-driven pathology” has satisfactorily addressed all previous comments and concerns. There are only a couple minor additional comments to consider:

1. Figure 3E is new microscopy data and would be more convincing with the inclusion of quantitative analysis

2. On p.8 line 252, the last word should be “form” rather than “from”.

Reviewer #3: (No Response)

**Part II – Major Issues: Key Experiments Required for Acceptance**

Reviewer #1: (No Response)

Reviewer #3: (No Response)

**Part III – Minor Issues: Editorial and Data Presentation Modifications**

Reviewer #1: (No Response)

Reviewer #3: (No Response)

PLOS authors have the option to publish the peer review history of their article (what does this mean?). If published, this will include your full peer review and any attached files.

Reviewer #1: No

Reviewer #3: No

Figure Files:

Data Requirements:

Reproducibility:

References:

---

## [Editor Report · Decision Letter 2]

29 Aug 2021

Dear Dr. Meunier,

We are pleased to inform you that your manuscript 'Host phospholipid peroxidation fuels ExoU-dependent cell necrosis and supports Pseudomonas aeruginosa-driven pathology' has been provisionally accepted for publication in PLOS Pathogens.

Best regards,

Gregory P Priebe, M.D.

Guest Editor

PLOS Pathogens

David Skurnik

Section Editor

PLOS Pathogens

Kasturi Haldar

Editor-in-Chief

PLOS Pathogens

orcid.org/0000-0001-5065-158X

Michael Malim

Editor-in-Chief

PLOS Pathogens

orcid.org/0000-0002-7699-2064
---

## [Editor Report · Acceptance letter]

8 Sep 2021

Dear Dr. Meunier,

We are delighted to inform you that your manuscript, "Host phospholipid peroxidation fuels ExoU-dependent cell necrosis and supports Pseudomonas aeruginosa-driven pathology," has been formally accepted for publication in PLOS Pathogens.

Best regards,

Kasturi Haldar

Editor-in-Chief

PLOS Pathogens

orcid.org/0000-0001-5065-158X

Michael Malim

Editor-in-Chief

PLOS Pathogens

orcid.org/0000-0002-7699-2064